

# From soil water to surface water – how the riparian zone controls the transport of major and trace elements from a boreal forest to a stream

Fredrik Lidman[1], Åsa Boily[1], Hjalmar Laudon[1], Stephan J. Köhler[2]

[1]Department of Forest Ecology and Management, Swedish University of Agricultural Sciences, Umeå, SE-901 83, Sweden
[2]Department of Aquatic Sciences and Assessment, Swedish University of Agricultural Sciences, Uppsala, P.O. Box 7050, SE-750 07, Sweden

*Correspondence to*: Fredrik Lidman (fredrik.lidman@slu.se)

**Abstract.** The riparian zone is the narrow strip of land, which lines lakes and watercourses. In the boreal region the riparian zone of headwaters often tends to be -wetland-like with high concentrations of organic matter, low pH and more or less reducing conditions. This means that riparian soils in many respects are different from the podzols and other types of mineral soils that dominate the boreal landscape. In this study a large number of major and trace elements (Al, As, B, Ba, Ca, Cd, Cl, Co, Cr, Cs, Cu, Fe, K, La, Li, Mg, Mn, Na, Ni, Pb, Rb, Se, Si, Sr, Th, Ti, U, V, Zn, Zr) and other parameters such as sulphate, total organic carbon (TOC) and pH were analysed in groundwater and soil water in a boreal hillslope. The objective was to investigate how the chemistry changes as the groundwater passes through the riparian zone and enters the stream.

Most of the investigated elements displayed substantially higher (up to 60 times) concentrations in the riparian soils than in the uphill mineral soils. There were also examples of nearly uniform concentrations throughout the transect (e.g. Na and Si), but in only a few cases were lower concentrations observed in the riparian zone (e.g. K, Mg and Ca). The degree of enrichment in the riparian soils could be linked to the affinity for organic matter, indicating that the pattern with strongly elevated concentrations in riparian soils is typical for organophilic elements. However, the elevated concentrations of many elements in the riparian zone could not be linked to increased uptake in biota (bilberry leaves and spruce shoots). In summary, the results confirmed that the riparian zone plays a central role for regulating the stream water quality and the transfer of a wide range of elements from terrestrial to aquatic ecosystems. Hence, riparian soils are essential for understanding the fate of nutrients, pollutants and other substances in the boreal landscape.





## 1 Introduction

The riparian zone is the interface between the terrestrial landscape and the streams and rivers draining it. Hence, the riparian soils are often the last material that the groundwater is in contact with before it becomes surface water. Although it normally occupies only a minor fraction of the total catchment area, it can therefore have a disproportionately high impact on the water quality of streams, rivers and lakes (Billett and Cresser, 1992). Similarly, it may also exert the ultimate control over the fluxes of many nutrients, pollutants and other substances from the terrestrial to the aquatic ecosystems due to its strategic location in the landscape. The importance of the concept of the riparian zone hinges on the fact that riparian soils can be profoundly different from the uphill soils that dominate the landscape (Sauer et al., 2007). In the boreal region shallower groundwater levels and generally wetter conditions tend to prevail in the vicinity of stream channels (Grabs et al., 2012). As a result, there is often an accumulation of organic matter, especially along small streams, which creates more wetland-like conditions in the riparian zone. Because of the importance of small streams for the runoff generation the presence of organic-rich riparian soils has far-reaching consequences for the water quality of boreal lakes and watercourses (Bishop et al., 2008; Lassaletta et al., 2010). The geochemistry of the riparian zone has been shown to be an important predictor for the stream water quality over wide spatial scales (1-1,000 km$^2$) (Smart et al., 2001). In the case of carbon the significance of the riparian zone is well-established both when it comes to dissolved organic carbon (DOC) and $CO_2$ (e.g. Fiebig et al., 1990; Hinton et al., 1998; Oquist et al., 2009; Lyon et al., 2011; Winterdahl et al., 2011; Grabs et al., 2012; Knorr, 2013; Fraser Leith, 2014; Ledesma et al., 2015). However, the special biogeochemical conditions of the riparian zone also have consequences for the transport of widely different substances such as nitrogen (Hill, 1996; Cirmo and McDonnell, 1997; Sabater, 2003), phosphorus (Mulholland, 1992), base cations (Ledesma et al., 2013), aluminium (Cory et al., 2007), mercury (Bishop et al., 2009), persistant organic pollutants (Bergknut et al., 2011) and pesticides (Vidon et al., 2010). Furthermore, the riparian zone is gaining increasing recognition as an ecological hotspot (Jansson et al., 2007; Kuglerová et al., 2013; Nilsson et al., 2013). As a consequence, precautions are often taken in modern forestry in order to reduce the deleterious impact of logging on riparian soils (Moore and Richardson, 2012; Kuglerová et al., 2014; Tiwari et al., 2016). Good arguments have also been put forward that the riparian zone is a key to understanding the temporal variability of stream water chemistry (Seibert et al., 2009).

However, for the vast majority of elements the role of the riparian zone remains unexplored. Yet, there are good reasons to hypothesise that the riparian soils influence the transport of many more substances than the above given examples. In the boreal region the mobility of many elements and miscellaneous hydrophobic substances is strongly connected to the presence of organic matter, both by accumulation in peat and other forms of solid organic matter (e.g. González et al., 2006; Cloy et al., 2009; Lidman et al., 2013) as well as by transport by organic colloids and particles (e.g. Gustafsson et al., 2000; Dahlqvist et al., 2007; Pokrovsky et al., 2010). The often elevated organic content of riparian soils in comparison to normal forest soils is also associated to fundamental chemical parameters such as pH and redox potential, which are well-known to



influence the speciation and mobility of many elements. The purpose of this study was therefore to broaden the perspective on riparian soils by investigating the behaviour of a wider range of elements as they are transported across the chemical gradients of the riparian zone in a boreal hillslope transect, starting in an uphill podzol profile and following the chemical evolution of the groundwater all the way to the stream. By using a multi-element approach, comprising 32 different elements

plus TOC, sulphate and pH, we hypothesised that it might be possible to identify some of the key processes that govern the behaviour of various substances in the riparian zone. In order to also capture the temporal variability in soil water and groundwater chemistry along the transect samples were collect at ten sampling occasions between January and October, covering the hydrologically most active period at the studied site. This approach resulted in a large amount of data so the intention here is not to discuss any of the investigated elements in particular detail, but rather to provide a general description

of how the water chemistry changes with respect to various types of elements as the water passes through the riparian zone. We believe that a better appreciation of the riparian zone is important for assessing both the water quality and long-term mass balance of the boreal landscape with respect to weathering products, pollutants, nutrients and other substances.

## 2 Material and methods

### 2.1 Site description

The investigated hillslope is located in the Krycklan catchment in northern Sweden (Laudon et al., 2013). It drains to a small first order stream (0.12 km$^2$), referred to as Västrabäcken or C2 in previous publications. The stream water sampling site is located ca. 300 m downstream of the investigated transect and is of the main sites included in the Krycklan Catchment Study (Fig. 1). The transect consists of three sampling sites located 4 m, 12 m and 22 m, respectively, from the stream as measured along the flow pathway of the groundwater. The perpendicular distance to the stream is considerably shorter than this, less

than 2 m for the closest site. The transect is sometimes referred to as the S transect in previous publications so the three sites are therefore called S4, S12 and S22, respectively. At each site suction lysimeters were installed in 1995, allowing regular sampling of soil water and groundwater from different depths in the three soil horizons. In this study four depths in S4 and five depths in S12 and S22, respectively, were sampled. The different lysimeters are referred to as S4, S12 and S22 followed by the depth in centimetres. For example, S22-90 means that the water was sampled 22 m from the stream at a depth of

90 cm. S22 is an iron podzol with clearly developed soil horizons. As such it is probably typical for most of the investigated catchment and in a broader sense also for boreal forests in general. At the surface there is a thin organic layer (mor), but apart from that the organic content is low (<0.8% below 10 cm) (Table S1). As the groundwater approaches the stream the podzols gradually give way for more organic histosols as a result of the increasingly wet conditions closer to the stream channel. S4 and S12 can both be considered to represent the riparian zone, while S22 in contrast represents a more ordinary

forest soil. The thickness of the organic layer increases from 20-30 cm in S12 to ca. 80 cm in S4 (Fig. S1). The porosity of the soils also varies (36-83%) with higher values in more organic layers and, consequently, a decline with depth in all profiles (Table 2). The continuation of the transect further upstream from S22 varies somewhat depending on the



hydrological conditions, but the distance to the water divide is ca. 100-120 m (Fig. 1). The transect has low relief with an inclination of ca. 3%, as compared to the average slope of the entire catchment (C2), which is 8.7%. The accumulation of organic matter is typical for the riparian zone of this catchment (Fig. S2).

The soils in the transect are made up by several meters of glacial till, but at a depth of ca. 1 m there is a compact layer of basal till with low hydraulic conductivity (Table S2; Bishop, 1991). Previous investigations of the hydrology in the transect have indicated that most of the water transport takes place in above this compact till layer (Rodhe, 1989; Laudon et al., 2004; Peralta-Tapia et al., 2014). At all three sampling sites along the transect (S4, S12 and S22) there is clear relationship between the groundwater levels and the discharge in the stream (Seibert et al., 2009). The discharge increases exponentially with

rising groundwater levels following the so-called transmissivity feedback mechanism, suggesting that much of the water transport takes place the uppermost saturated soil layers  (Bishop et al., 2011).

The mineralogy is dominated by quartz (40%), K-feldspar (25%), plagioclase (23%), amphibolites (7%), muscovite (4%) and chlorite (1%) (Ledesma et al., 2013). The transect as well as the catchment as a whole is covered by century-old

coniferous forest, mainly Norway spruce *(Picea abies)* and Scots pine (*Pinus sylvestris*), with elements of deciduous trees and shrubs. The climate is cold temperate with an mean annual temperature of $1.8^{\circ}$C (1981-2010), ranging from $-9.5^{\circ}$C in January to $14.7^{\circ}$C in July (Laudon et al., 2013). The mean annual precipitation is 614 mm, of which roughly half is lost by evapotranspiration and half by runoff. The snow cover remains for 168 days per year on average (1980-2007), varying in maximum depth between 43 cm and 113 cm. The maximum soil frost depth varies between 2.5 cm and 79 cm (1993-2007)

depending on factors such as temperature and the timing and depth of the snow (Haei et al., 2010).

The stream channel was deepened, partially straightened and probably also extended further up in the catchment by manual ditching in the 1920s. Ditching of forests was a common practise in Fennoscandia  at this time, and a large portion of the Swedish headwaters have therefore been affected in a similar way (Dahlstrom et al., 2005). The purpose was to improve the

drainage, thereby counteracting the paludification and increasing the forest productivity. Exposed roots of some older trees in the riparian zone testify that there may have been a lowering of the ground level as a result of the ditching. Presumably peat, which had been building up under the previously wetter conditions, has begun to decay, causing the ground surface to sink by ca. 2-3 dm in some places. Further details on the transect can  be found elsewhere (Nyberg et al., 2001; Stähli et al., 2001).

**2.2 Sampling and analyses**

Soil water and groundwater was collected from the lysimeters at ten occasions during 2008. The investigation period started during winter baseflow conditions in January and ended in October, when the system again had returned to winter baseflow conditions. Samples were collected every month with the exception of June, when two samples were collected, and




September, which was omitted. This sampling strategy was assumed to capture the most active period both in hydrological and biogeochemical terms. This means that there are in most cases ten observations from each depth in each profile, but in certain cases there are values missing either because sufficient amounts of water could not be collected due to soil frost or draught or because the concentrations of some elements in some cases were below detection limit.

In S4 four lysimeters were sampled, reaching from 35-65 cm in depth. This  cover the most hydrologically dynamic part of the profile, since S4-35 never was saturated, except possibly briefly in connection with the peak flow of the spring flood, and S4-65 was constantly saturated, except for a few weeks in the middle of the summer. In S12 five lysimeters covering a depth from 20-70 cm were sampled. S12-20 is believed never to have been saturated, while the groundwater level occasionally may have fallen below S12-70 during the driest parts of the summer. In S22 five lysimeters reaching from 20-90 cm were

sampled. Except in connection with the peak flow of the spring flood S22-20 is expected to have been above the groundwater table, whereas S22-75 and S22-90 are expected to have been in the saturated zone throughout the entire sampling period. The groundwater tables at the three sampling sites S4, S12 and S22 were reconstructed using continuous discharge measurements from the nearby downstream stream location (Laudon et al., 2004). With slightly revised regressions the $R^2$ values were 0.96 (S4), 0.94 (S12) and 0.87 (S22), respectively (p<0.001). Discharge was measured ca.

300 m downstream from the transect at V-notch weir using a pressure transducer connected to a data logger (Fig. 1; Seibert et al., 2009).

The chemical analyses included Cl, sulphate, TOC, Al, As, B, Ba, Be, Ca, Cd, REEs, Co, Cr, Cs, Cu, Fe, K, Li, Mg, Mn, Na, Ni, Pb, Rb, Se, Si, Sr, Th, Ti, Tl, U, V, Zn and Zr. Because of the large similarities between the REEs only La is presented in

this study in order to save space and make graphs and tables clearer and more manageable. The fractionation within the lanthanide series will for this reason have to be addressed elsewhere. pH was not directly measured in the water samples from 2008, but older measurements (1996-1998) have established a strong relationship between pH on one hand and TOC and Ca on the other (Fig. S3), which was used to estimate the pH of the analysed water samples in this study.  The average prediction error was 0.24 pH units. Additional analyses of water from the adjacent stream (referred to as C2 or Västrabäcken

in previous studies) were taken from the material presented by Lidman et al. (2014).

The soil water and groundwater was sampled by attaching vacuum bottles to the lysimeters for 2-3 days. The lysimeters were carefully installed in 1995 so any effects of disturbance on the soils were expected to have evanesced. The samples were filtered (0.45 µm) and acidified to pH<2 with ultrapure double-distilled $HNO_3$. TOC was then analysed by a Shimadzu TOC-

VCPH instrument. Previous research in Krycklan has demonstrated that the amount of particulate organic carbon rarely exceeds a few percent in these systems so the total organic carbon (TOC) is essentially equal to the dissolved organic carbon (DOC) (Laudon et al., 2011). Anions were measured by ion chromatography (Dionex ICS-90, Sunnyvale, Ca, USA; 4 mm i.d. AG14 and AS14 columns) using a suppressor and conductivity detector. All other elements were analysed by ICP-MS (Perkin-Elmer ELAN 6000).





The speciation of elements in the riparian soil water was calculated using thermodynamic modelling in Visual MINTEQ 3.1 (Gustafsson, 2012). The binding to DOC was modelled using the Stockholm Humic Model as described by Sjostedt et al. (2010). Following the definition suggested by Lidman et al. (2014) the modelled association to DOC was used as an index for the affinity for organic matter ($\Omega$). In a regression analysis $\Omega$ was transformed using the logit function:

$$logit(\Omega) = log\left(\frac{\Omega}{1-\Omega}\right) \tag{1}$$

All statistical analyses, including the principal component analysis (PCA), were made in R (R Core Team, 2014). In order to remove all missing values Cr and Th were excluded from the PCA of the soil water and groundwater and one value each for U and Zr were interpolated from the two surrounding lysimeters (Table 1).

The biological uptake in the riparian zone and in the upslope soil, respectively, was tested by collecting fresh bilberry leaves (*Vaccinium myrtillus*) and spruce shoots (*Picea abies*) early in the growing season (11 June 2013). Two grouped samples of each species were collected in the vicinity of the lysimeters at S4 and S22, respectively. A screening of the element concentrations were made using ICP-MS at a commercial laboratory (ALS Global, Luleå, Sweden) following certified standard procedures. We report the results for the 19 elements, which were measured with an accuracy of two of more significant digits.

## 3 Results and discussion

### 3.1 Soil water and groundwater chemistry

For most elements there was a substantial increase in the concentrations along the transect, from the upslope profile (S22) to the riparian profile (S4) (Table 1, Table S3). These patterns, which were persistent throughout the year, are exemplified by TOC and V in Fig. 2. In the case of TOC the increase in riparian soil water and groundwater was not surprising given that the organic content of the riparian soils are substantially higher (up to 33 mg L$^{-1}$ on average) than the upslope soils, which typically contained only a few mg L$^{-1}$ (Fig. S1). This is a trend, which appears to be typical for the riparian soils of many small boreal streams (Grabs et al., 2012). V followed that same trend as TOC with substantially higher (nearly two orders of magnitude) concentrations in the riparian zone. However, there were also elements, whose concentrations did not increase markedly in the riparian zone. One such example is Na, which occurred in relatively stable concentrations around 2 mg L$^{-1}$ throughout the entire transect. Further examples (Ca, Al, La, K, Fe and Mn) can be found in Figs. S6 and S7, where Al, La, Fe and Mn all exhibit similar patterns as TOC and V. In the case of Al, this confirmed previous studies in this transect (Cory et al., 2007).

pH was estimated to be on average 4.5 in S4, ranging from 4.2-4.7 with slightly higher values in deeper horizons: on average 4.3 in S4-35 as compared to 4.6 in S4-64. In S12 the pH was on average 5.0, covering an interval from 4.4 to 5.2. Again





there was a tendency of higher pH in deeper horizons. The average pH in S22 was only slightly higher than in S12, 5.1, but there was a notable difference of more than one pH unit between S22-90, which had an average pH of 5.9, and the upper horizons, which had an average of 4.8. Due to the stable pH in all profiles of S22 this also coincides with the observed range in pH, 4.8-5.9. Hence, the pH gradient experienced by substances transported from the uphill podzol to the riparian zone is

approximately in the order of 0.5-1 pH units. This shift in pH alone may, however, have palpable consequences for the speciation of many metals and the solubility of solid phases such as Al(OH)3 and various Fe precipitates (Gustafsson, 2001; Sauer et al., 2007; Sjostedt et al., 2010), and even more so when combined with the increase in TOC concentration (Fig. 2). Detailed studies of the Fe speciation using X-ray absorption spectroscopy (XAS) have revealed considerable complexity with simultaneous presence of both $Fe^{2+}$ and $Fe^{3+}$ throughout the transect (Sundman et al., 2014). In S4 the aqueous Fe

speciation was completely dominated by organic Fe complexes, which suggests that that the riparian zone is responsible for the absence of the Fe colloids in the adjacent stream (Neubauer et al., 2013). Normally Fe colloids are an important vector for many major and trace elements in boreal rivers (Gustafsson et al., 2000; Dahlqvist et al., 2007), but while such Fe precipitates should be present in the upslope podzols the lower pH and higher TOC concentration of the riparian zone apparently destabilised or blocked these precipitates from reaching the stream water. Only as the significance of the riparian

zone decreased with increasing catchment area, e.g. due to less organic riparian zones and a larger influence of deep groundwater (Peralta-Tapia et al., 2015; Lidman et al., 2016), will these Fe colloids start to appear in the stream water (Neubauer et al., 2013).

## 3.2 Principal component analysis

In order to further explore the trends in the soil water and groundwater chemistry a principal component analysis (PCA) was

conducted (Fig. 3). There was a considerable variance in the dataset, and as much as nine principal components were needed to explain more than 95% of it. The first two principal components could explain in total 57% of the variance with PC1 contributing with 33% and PC2 with 24%. A noteworthy pattern in Fig. 3 is that most lysimeters from the same profile fall close to one another. This suggests that the variance in the transect mainly is horizontal rather than vertical. This underlines the relevance of the concept of the riparian zone, since it demonstrates that the riparian profile (S4) is decisively different

from the upslope mineral soil (S22). The only deviations from the division into S4, S12 and S22, respectively, were the deepest lysimeter in S22 (S22-90) and the shallowest lysimeter in S12 (S12-20). In the case of the former, there was a notable decrease in the hydraulic conductivity below ca. 80 cm in S22 (Table S2), suggesting that the exchange between S22-90 and shallower soil layers is limited. This lysimeter was for instance characterized by high concentrations of base cations, particularly K, Ca and Mg, and higher pH (Figs. S6 and S7; Table 1). It has been demonstrated that the

concentration of base cations in the area tends to increase with the age of the groundwater (Klaminder et al., 2011), which supports the idea that S22-90 represents a less hydrologically active layer. As concerns S12-20, it represents a more organic soil than the deeper horizons of S12 (Fig. S1), which for instance causes it to have a higher TOC concentration (Fig. 2; Table 1).



Most of the elements are found on the same side of the biplot as the S4 lysimeters (e.g. Ba, Cd, Al, Zr and Ni), indicating that they occurred in higher concentrations in the riparian zone. However, some of them (e.g. Li, Co, Si, Cu and Tl) are shifted towards the S12 lysimeters, while others (e.g. Pb, Fe, As, Zn, TOC, Ti and V) are shifted slightly towards the S22 lysimeters. A strong association to the S12 lysimeters is mainly displayed by Mn and Sr, signifying that they occurred in higher concentrations in S12. The S22 lysimeters are surrounded by the alkali metals K, Rb and Cs, but also B and Se. In the case of Se the higher concentrations in S22 may be related to the more oxidising conditions, since the solubility of Se easily decreases in more reducing environments (Gustafsson and Johnsson, 1992). The special patterns of Se in the soil water and groundwater are also consistent with the dynamic behaviour of Se concentrations in adjacent stream (Lidman et al., 2011). Finally, some major ions like Ca, Mg, Na and sulphate are found in between the S12 and the S22 lysimeters in the biplot.

### 3.3 Enrichment in the riparian zone

The grouping of the lysimeters at the same distance from the stream suggests that it is possible to reduce some of the complexity of the dataset by investigating the average concentrations at each of the sites. Since the principal change in the soil water and groundwater chemistry occurred longitudinally along the transect, a comparison of the average concentrations in the riparian zone (S4) to the average concentrations in the upslope mineral soil (S22) can give a fair overview of the major trend in the transect. It has already been observed that many elements occurred in substantially higher concentrations in the riparian zone (Table 1, Fig. 2), but when plotting the ratio of the average concentration in S4 and the average concentration in S22 it becomes evident just how different the riparian zone is from the uphill mineral soil (Fig. 4). For the vast majority of elements the concentrations were considerably higher in the riparian zone. A few elements – most notably K, but also Ca, Mg and Se – showed a contrasting pattern with higher concentrations in the uphill soils, while yet a few others, e.g. Rb, Na and sulphate, occurred in similar concentrations throughout the transect. However, a closer look at the observations (Table 1) shows that the higher average concentrations of K, Ca and Mg in the mineral soil (S22) can be attributed to the deepest lysimeter in S22, which also stood out in the PCA (Fig. 3). Thus, excluding this sample from the averages will remove most of the differences between S4 and S22 for K, Ca and Mg.

Most elements, however, displayed considerably higher concentrations in the riparian zone than in the uphill mineral soil. The far right side of Fig. 4 is occupied by metals, which are well-known for their low solubility, e.g. Zr, Th and Al. The most extreme example is V with more than 60 times higher concentrations in the riparian zone than in the mineral soils further up in the transect. Unlike Zr, Th and Al the speciation of V is directly affected by the redox conditions, which may contribute to the differences in its concentrations (Agnieszka and Barbara, 2012). However, it should be noted that the Th concentrations in S22 fell below the detection limit in all but one of the lysimeters. This was the shallowest lysimeter (at 20 cm) with the highest TOC concentration. Therefore, there are good reasons to believe that the enrichment factor (S4/S22) for Th and also Cr were underestimated. If it is assumed that the Th concentration in S22 is correlated to the TOC concentration or that it



follows the same pattern as Zr and V, this would be enough to push the S4/S22 ratio well above 100. However, which element is most enriched and to what degree is perhaps not a crucial issue. More important is to identify what chemical properties cause some elements to occur in elevated concentrations in the riparian soil water and groundwater and others not to.

The general impression from both the PCA (Fig. 3) and, more specifically, the S4/S22 ratio (Fig. 4) is that more insoluble or immobile elements tend to occur in higher relative concentrations in the riparian zone, while generally more mobile elements such as the base cations and anions like Cl and sulphate occur in more uniform concentrations throughout the transect. In boreal waters more immobile elements are often to a large extent transported by colloids, typically Fe or organic colloids,
which tend to represent two different colloidal populations (Lyven et al., 2003). As discussed above, the Fe colloids appear to be missing in the riparian zone, leaving only the organic colloids as possible carriers (Neubauer et al., 2013). There is also strong evidence that organic matter is responsible for carrying a wide range of elements in boreal waters (e.g Gustafsson et al., 2000; Pokrovsky et al., 2005; Andersson et al., 2006; Dahlqvist et al., 2007; Pokrovsky et al., 2010; Vasyukova et al., 2010; Aiken et al., 2011). Furthermore, it has been demonstrated that the affinity for organic matter of various metals is
essential for understanding their transport and spatial variability in the boreal landscape (Lidman et al., 2014). Using the same definition of the affinity for organic matter as was proposed by Lidman et al. (2014), makes it possible to quantitatively test the hypothesis that the enrichment in the riparian soil water and groundwater was related to the affinity for organic matter. Figure 5 shows the S4/S22 ratio, i.e. the enrichment in the riparian zone (S4) as compared to the upslope mineral soil (S22), as function of the affinity for organic matter. The affinity for organic matter is expressed in percent and represents the
proportion of each element that is expected to be bound to DOC in the soil water and groundwater of S4, but in this application it was used mainly as an index for the strength of the affinity for organic matter.

The selection of elements in Fig. 5 was based on the analysed elements and the available thermodynamic constants in Visual MINTEQ 3.1 (Gustafsson, 2012). Some elements like Fe and Mn were excluded because uncertainties regarding their
oxidation state make the affinity for organic matter ill-defined. U was, however, included as $U^{6+}$ despite its redox chemistry, since any presence of $U^{4+}$ just would increase the already high affinity for organic matter of $U^{6+}$ (>96 %) slightly. As can be seen in Fig. 5, where the affinity for organic matter has been transformed using the logit function (Equation 1), there was a strong positive correlation between the affinity for organic matter and the enrichment in the riparian soil water and groundwater (r=0.88 , p<0.01). The modelling also indicated that many of the elements, which are enriched in the riparian
zone, would not be soluble in such high concentrations had it not been for the high TOC concentrations. Yet, Fig. 5 cannot be taken as direct evidence that each of the enriched elements in the riparian zone is governed primarily by TOC as there also may be other controlling factors, especially in the uphill podzol profile (S22), where the TOC concentration was low. Yet, our data strongly suggest that TOC is a crucial parameter for understanding the role of the riparian soils.





Although it is beyond the scope of this work to analyse the time series for all the investigated elements in the lysimeters, it is illustrative to look at a few examples that can shed further light on the role of organic matter in riparian soils. Ti is one example of a relatively insoluble metal, which should have a low mobility unless there is some carrier phase such as TOC present. Figure S4 shows all of the observations of Ti plotted against TOC, revealing a strong positive correlation (r=0.97, p<0.001) between Ti and TOC for all soil water samples. Clearly, high Ti concentrations were encountered solely in samples with high TOC concentrations.

The significance of TOC was also supported by the temporal variability of organophilic elements in the soil water and groundwater, at least in the more organic soils. One example is Al, which in each of the lysimeters in S4 had a significant (p>0.05) positive correlation (r>0.84) to TOC when looking throughout the entire sampling period (see details in Fig. S5). At times when TOC increased so did Al, supporting the hypothesis that the mobilisation and transport of organophilic elements in the riparian zone largely could be controlled by TOC. Similar patterns were, however, absent in S22 so the mobility of Al in these inorganic soils was evidently controlled by other processes. For instance, it has been shown that the mobility of Al in the Bs horizon of podzol soils can be limited by the precipitation of $Al(OH)_3$ and in some cases imogolite (Gustafsson, 2001). Hence, while the large differences in TOC concentrations between the uphill mineral soils and the riparian zone were responsible for the elevated levels of many elements in the riparian soil water, this does not necessarily imply that it also controls their mobility in each of the investigated soil layers.

## 3.4 Influence on the stream water chemistry

The analyses of the soil water and groundwater along the investigated transect demonstrated that the riparian zone can induce considerable changes in the water chemistry. It is therefore pertinent to ask what influence the riparian zone has on the stream water quality. Had there been no differences between the riparian zone and the uphill mineral soils one should expect that the stream water would resemble the groundwater in S22. However, since the groundwater changes character as it approaches the stream and passes through the organic riparian soil, one might expect that stream water chemistry instead would resemble the riparian groundwater. As discussed above, the riparian zone is that last terrestrial environment that the groundwater is in contact with before it enters the stream so it could potentially have a disproportionally strong impact on the stream water composition.

When comparing the uphill mineral soil (S22) to the stream (C2), many elements (e.g. Th, Al, U, Fe and La) occurred in considerably higher concentrations (up to 60 times) in the stream water than in the soil water (Fig. 6). Yet, stream water concentrations were missing for some of the metals, e.g. V and Zr, which showed the largest enrichment in the riparian soil water. The difference between the stream water and the uphill soil water is remarkable given that this podzol profile is likely to be reasonably representative for the vast majority of soils throughout the catchment. For example, rapid degradation of TOC and subsequent precipitation of Fe, Al and associated elements are typical processes in the development of podzols





(Gustafsson, 2001; Sauer et al., 2007; Sundman et al., 2014). This illustrates that the riparian zone has a profound impact also on the stream water chemistry for a wide range of elements. Based on Fig. 5 the effects on the stream water chemistry seem to mainly concern elements with a high affinity for organic matter.

When comparing the riparian soil water (S4) to the steam water (C2), the concentrations were in many cases even higher in the riparian soil than in the stream (up to 4-5 times). In other words there was no perfect match between the riparian soil water and the stream water (Fig. 6). However, the riparian soil water in this study represents only a single transect, whereas the streams water displays an integrated signal for the entire catchment. Although the overall appearance and functioning of the investigated transect should be representative for the catchment at large, there is also a longitudinal variation in the composition of the riparian zone along the stream. For example, the riparian zone of the investigated transect represented an area with higher organic content (around the 3$^{rd}$ quartile) than most reaches of the stream (Figs. S1 and S2). If the interpretation that the organic matter is controlling the transport of these elements is correct, that could explain why the studied riparian zone exhibited higher concentrations than the stream. Differences in the organic content of the riparian zone along the stream is known to derive from variations in the topography and the groundwater flow pathways (Tiwari et al., 2016). This hypothesis is also supported by the fact that the TOC concentrations in the riparian zone also were higher than in the stream (Fig. 6).

There were only three elements, Ca, Mg and K, which displayed the opposite pattern: higher concentrations in the stream water (C2) than in the riparian soil water (S4) (Fig. 6). These three elements also stood out in the measurements in the transect because they occurred in higher concentrations in the mineral soil (S22) than in the riparian soil (S4), mainly due to the high concentrations in the deepest lysimeter in S22 (Fig. 4). In the case of Ca and Mg, the agreement between the mineral soil water (S22) and the stream water (C2) was good, whereas the K concentration in the stream water was more than twice as high as in the mineral soil water. K and, possibly, Rb were the only examples of elements with notably higher concentrations in the stream water than in the three soil profiles. The higher concentrations of base cations like K, Ca and Mg in the stream water could indicate a contribution of deeper groundwater to the stream, but at the same time related weathering-products such as Sr, Na, Rb, Cs and Si all showed relatively uniform concentrations throughout the transect and the stream. It is possible that the differences again are cause by longitudinal heterogeneities along the stream.

### 3.5 Biological uptake

It has been shown that the species richness of the riparian zone is higher along stream reaches with high groundwater discharge. Hence, the transport of nutrients to and through the riparian zone evidently has ecological consequences (Kuglerová et al., 2013). The results of this study have demonstrated that organic soils in the riparian zone also can have nearly two orders of magnitude higher concentrations of certain elements than normal forest soils. It is therefore pertinent to





ask whether the riparian zone also is a hotspot for biological uptake of natural, but undesirable, elements or anthropogenic pollutants.

The screening of bilberry leaves from the riparian forest (near S4) and the uphill forest (near S22), respectively, showed no clear relationship between the concentrations in soil water and the concentrations in biota (Table S4; Fig. 7). In other words, the elevated concentrations of many elements in the riparian zone did no lead to higher concentration in bilberry leaves. The results were similar for spruce shoots (Fig. S8). Also when considering the more unreliable results for elements, which were only reported with one significant digit (e.g. Sb, Pb, Cr, La, Ti, V and Zr), there were no clear signs of elevated concentrations in riparian biota. (Possible exceptions were, however, As, Co and Mo.) Combined these results suggest that the biological uptake either in general is actively regulated by the plants or that the elements are present as species with low bioavailability. The two possibilities do not exclude each other and may vary from element to element, but according to the discussion above and the modelling of the chemical speciation one should expect a large portion of the elements with elevated concentrations in the riparian soil water to be associated with TOC (Fig. 5). For a wide range of metals it has been shown that their bioavailability tends to decrease when they are bound to organic matter, e.g. Pb (Van Sprang et al., 2015), U (Trenfield et al., 2011a; Trenfield et al., 2011b), Ni (Weng et al., 2004; Deleebeeck et al., 2009), Cu (Deruytter et al., 2014) and Zn (De Schamphelaere et al., 2005). Although there may be differences between species, the general effects on the bioavailability are likely to be similar also for spruce, bilberry and many other plants. Hence, the fairly constant concentrations in biota could be explained by the fact that the concentrations of the most bioavailable chemical species do not differ much between the inorganic uphill soils and the riparian zone. Instead, the enrichment occurs mainly by increasing the amount of the organically bound fractions, which have low bioavailability.

Considering all elements in both bilberry leaves and spruce shoots there was, however, one notable exception, displaying a clear enrichment in riparian biota, namely Cs. The soil water concentration of Cs in the riparian zone was 2-3 times higher than in the uphill soil, and the concentrations in biota were 4-5 times higher. Thus, Cs was the only element, which exhibited more than twice as high concentrations in riparian bilberry leaves (and spruce shoots) as in the uphill ditto. Since Cs is considered to have low affinity for organic matter, the general explanation that the elevated element concentrations in the riparian soil water are caused by higher TOC concentrations is unsatisfactory for Cs (Fan et al., 2014). It seems more likely that factors such as lower amount of mineral surfaces and lower pH might be the reason behind the higher Cs concentration in the riparian zone. In any case, the additional Cs found in the riparian soil water is unlikely to be associated to TOC. Instead, Cs is likely to largely be present as free Cs ions in the riparian zone so no effects on the bioavailability from TOC can be expected in this case. There was also one notable example of the opposite, namely lower concentrations in riparian biota. This exception was Mn, which possibly could be related to its redox behaviour, causing Mn to be more present as $Mn^{2+}$ in the riparian zone (Fig. 7).





### 3.6 The importance of the riparian zone

The perhaps most important observation in this study was the large differences between the riparian soils and the uphill forest soils for many elements. As discussed above, these differences have previously been known for a few elements, but judging from this study the enrichment in the water of the riparian zone appears to be a rule more than an exception (Fig. 4).

The degree of enrichment in the riparian soil water and groundwater could be linked to the affinity for organic matter (Fig. 5), which suggest that similar enrichment should be expected for many other metals and organophilic substances in general, e.g. organic pollutants (Bergknut et al., 2011).

Since this study focused on a single hillslope transect, a key question is unavoidably how representative the results are.

Needless to say, factors such as topography, hydrology, soil composition etc. will vary from catchment to catchment so there is little reason to expect the exact numbers, e.g. the degree of enrichment of organophilic elements in riparian soils, to repeat themselves. On a more conceptual level, however, the gradient represented by this transect from relatively dry organic-poor mineral soils in uphill locations to wetter organic soils near the streams is surely not untypical for the boreal landscape (Sauer et al., 2007; Grabs et al., 2012). Despite their size, small streams and headwaters are fundamental for generating

runoff in the boreal landscape (Bishop et al., 2008). If the water quality of these streams and the transfer of TOC and associated elements are controlled by riparian soils, as supported by this study, there are also good reasons to believe that the effects of the riparian soils have implications at much larger scales than the investigated transect (Laudon et al., 2011; Ledesma et al., 2015). The results clearly demonstrated that the riparian zone had a profound impact on the stream draining the transect (Fig. 6), and previous research in the Krycklan catchment has shown that the water chemistry of this stream (C2)

in many respects is typical for forested catchments in the area despite its small size (Cory et al., 2006; Björkvald et al., 2008; Lidman et al., 2012; Köhler et al., 2014; Lidman et al., 2014). While riparian soils like the one in S4 are responsible for the high TOC concentrations and the low pH in small streams, they also destabilise many Fe compounds, thereby affecting the balance between organic colloids and Fe colloids, which are the two main carriers in boreal waters (Lyven et al., 2003; Neubauer et al., 2013; Köhler et al., 2014; Ledesma et al., 2015).

Arguably, the chemistry of the stream draining the investigated transect did not agree completely with either the water of the riparian zone or that of the uphill mineral soil (Fig. 6). However, given the heterogeneity and complexity of the natural landscape it is doubtful whether it at all would be possible to find something like a representative transect, even in a relatively small catchment like this one ($0.12 \ km^2$). In all catchments there is likely to be substantial longitudinal and

transverse variation in state factors such as topography, grain size distribution and mineralogy, which in turn gives rise to a heterogeneity in hydrology and biogeochemistry (Ledesma et al., 2015). Nevertheless, it was clear that the composition of the stream water had been significantly influenced by the riparian zone, e.g. by gaining substantially higher concentrations of TOC and organophilic elements in general (Fig. 4). For an element like C, which ultimately is derived from the atmosphere,





it is not hard to see how riparian soils can become a major source of TOC to the streams (Fiebig et al., 1990; Agren et al., 2008; Lyon et al., 2011). More intriguing are the associated elements, which in a similar manner as TOC have a major source in the riparian zone. These elements, however, are largely derived from weathering of local soils, but it is unclear where this weathering took place and what the long-term element budget for the riparian zone looks like. One possible

explanation would be that there is a preferential net weathering in the riparian zone. This could for instance be related to the higher TOC concentrations and the lower pH as compared to the uphill soils (Erlandsson et al., 2016). This would imply that much of the weathering of organophilic metals is limited to a small fraction of the landscape. Another possibility is that these elements initially were released mainly in the upslope mineral soils, which cover much larger areas than the riparian soils, and then have been transported to the riparian zone. Discussing the sources of Al in this transect, Cory et al. (2007)

hypothesised that there could be a flushing from uphill mineral soils to the riparian zone in connection with hydrological episodes such as the spring flood. The results of this study do, however, not support that hypothesis. The Al concentrations in the riparian zone (S4 and S12) were at all sampling occasions throughout the year substantially higher than in the uphill mineral soil (S22) so it seems unavoidable to draw the conclusion that the riparian zone currently is a net source for Al (and other organophilic elements) during all parts of the year. Hence, if one wishes to stick with the hypothesis that the vast part

of the weathering of these elements has occurred throughout the catchment, one has to assume that most of the organophilic metals, which now are being released from the riparian zone, were brought there and somehow accumulated during some earlier stage, when the riparian zone acted as a sink. For instance, it is possible that organophilic metals historically accumulated in the organic matter of the riparian zone in the same way as they still do in boreal mires (Lidman et al., 2013; Lidman et al., 2014). However, as a result of some change in the system, e.g. the ditching of the stream in the early 20[th]

century, these metals are now being released from the solid organic matter as it decomposes. Under this scenario the current stream chemistry of this and presumably many of all the other ditched streams would represent a historical anomaly. Which explanation is true – preferential weathering in the riparian zone or historical accumulation in combination with present mobilisation – seems precarious to answer based on only on the water chemistry throughout the transect. Instead this will require a more detailed investigation of the solid phase as well. A better understanding of the mass-balance of the riparian

zone would further elucidate how various substances – nutrients as well as pollutants – are transported from boreal forests to surface waters and help assess the effects of human perturbations such as logging, ditching and climate change on the quality of boreal waters.

## 4 Conclusions

The results of this study demonstrated that riparian soils have a profound effect on the element transport from a boreal forest

hillslope to the draining stream, particularly when it comes to elements with a high affinity for organic matter. For some elements the concentrations in the riparian zone were nearly 100 times higher than in the mineral soils less than 20 m upslope. It could be demonstrated that the degree of enrichment was related to the affinity for organic matter. Whereas



elements with low affinity for organic matter occurred in fairly uniform concentrations throughout the transect, elements with a high affinity for organic matter were strongly enriched. This enrichment also had implications for the stream water, which tended to have higher concentrations of DOC and organophilic elements than the soil water and groundwater of uphill soils. As a consequence, it can be expected that the water quality of boreal surface waters, especially in smaller catchments

and with respect to TOC and organophilic elements, can be quite different from that of the groundwater and soil water throughout most the catchments. This emphasises the importance of the riparian zone for understanding the water quality of the boreal landscape and the transfer of solutes from terrestrial to aquatic ecosystems. Despite the strong enrichment of organophilic elements in the riparian soil water no signs of elevated biological uptake in the riparian zone were observed. At present the riparian zone constitutes a source for a wide range of elements, but further studies are needed to determine the

long-term mass-balance of the riparian zone and the underlying causes of the enrichment. The results of this study emphasise that riparian zone is crucial for understanding the long-term fluxes of nutrients and pollutants in the boreal landscape as well as the historical and future water quality of boreal surface waters in the wake of a changing climate and modern forestry.

**Data availability**

The data will be made available through the homepage of the Krycklan Catchment Study (www.slu.se/Krycklan).

**Competing interests**

FL and ÅB were funded by the Swedish Nuclear Fuel and Waste Management Company (SKB).

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



**Table 1. Average element concentrations at different depths in the three investigated soil profiles. All concentrations are given in µg L$^{-1}$ except for TOC, which is given in mg L$^{-1}$. The number of samples included in the averages varies from 1-10, depending on availability of soil water, detection limits etc. See Table S3 for corresponding relative standard deviations.**

| depth (cm) | S4 | | | | S12 | | | | | S22 | | | | |
|---|---|---|---|---|---|---|---|---|---|---|---|---|---|---|
| | 35 | 45 | 55 | 65 | 20 | 30 | 40 | 60 | 70 | 20 | 35 | 50 | 75 | 90 |
| Al | 1700 | 1400 | 1200 | 1400 | 1200 | 640 | 520 | 410 | 330 | 68 | 1.6 | 13 | 8.4 | 50 |
| As | 0.95 | 0.99 | 0.83 | 0.60 | 0.88 | 0.37 | 0.22 | 0.22 | 0.37 | 0.057 | 0.0026 | 0.19 | 0.17 | 0.027 |
| B | 6.3 | 5.2 | 4.5 | 5.1 | 8.6 | 5.0 | 4.3 | 4.9 | 4.5 | 4.2 | 3.9 | 3.9 | 4.4 | 3.6 |
| Ba | 35 | 25 | 25 | 26 | 16 | 16 | 15 | 11 | 10 | 13 | 6.0 | 5.9 | 6.5 | 7.8 |
| Be | 0.20 | 0.16 | 0.15 | 0.21 | 0.17 | 0.11 | 0.091 | 0.099 | 0.11 | 0.0098 | 0.0047 | 0.0087 | 0.0080 | 0.0078 |
| Ca | 900 | 1100 | 1300 | 1100 | 810 | 1600 | 1800 | 2000 | 2100 | 1100 | 920 | 1000 | 1000 | 3500 |
| Cd | 0.12 | 0.16 | 0.12 | 0.13 | 0.061 | 0.044 | 0.046 | 0.070 | 0.096 | 0.016 | 0.012 | 0.010 | 0.017 | 0.0092 |
| Cl | 1.0 | 1.1 | 1.0 | 1.1 | 0.99 | 0.97 | 0.96 | 0.97 | 0.96 | 0.51 | 0.72 | 0.94 | 0.87 | 0.79 |
| Co | 1.4 | 1.5 | 1.3 | 1.5 | 0.70 | 0.94 | 1.1 | 1.1 | 1.4 | 0.23 | 0.11 | 0.17 | 0.14 | 0.027 |
| Cr | 2.2 | 1.7 | 1.4 | 1.6 | 1.7 | 0.75 | 0.54 | 0.44 | 0.40 | 0.25 | n/a | 0.15 | n/a | n/a |
| Cs | 0.017 | 0.022 | 0.039 | 0.015 | 0.075 | 0.019 | 0.013 | 0.011 | 0.013 | 0.012 | 0.0054 | 0.0074 | 0.0078 | 0.010 |
| Cu | 2.3 | 4.0 | 2.6 | 4.5 | 2.6 | 2.7 | 1.8 | 3.9 | 3.1 | 0.84 | 0.77 | 0.65 | 0.51 | 0.82 |
| Fe | 1400 | 1500 | 1300 | 740 | 370 | 110 | 57 | 55 | 160 | 5.3 | 74 | 24 | 42 | 43 |
| K | 110 | 120 | 170 | 100 | 480 | 190 | 150 | 250 | 370 | 360 | 120 | 210 | 210 | 1000 |
| Li | 3.6 | 3.2 | 3.2 | 3.5 | 2.6 | 2.5 | 2.7 | 2.7 | 2.6 | 0.52 | 0.48 | 0.58 | 1.4 | 0.67 |
| La | 2.2 | 1.7 | 1.6 | 1.5 | 1.3 | 0.87 | 0.73 | 1.4 | 1.6 | 0.24 | 0.013 | 0.088 | 0.043 | 0.17 |
| Mg | 300 | 380 | 500 | 350 | 420 | 500 | 500 | 590 | 590 | 320 | 310 | 360 | 340 | 1100 |
| Mn | 6.4 | 9.5 | 13 | 8.7 | 7.4 | 8.0 | 7..9 | 15 | 21 | 0.38 | 0.39 | 0.22 | 0.25 | 3.9 |
| Na | 2200 | 2000 | 2100 | 2000 | 1900 | 2000 | 1900 | 2100 | 2200 | 1500 | 1700 | 1900 | 1900 | 2400 |
| Ni | 3.0 | 3.9 | 3.5 | 3.7 | 1.2 | 1.5 | 1.7 | 1.9 | 2.5 | 0.83 | 0.23 | 0.29 | 0.30 | 0.80 |
| Pb | 0.67 | 0.28 | 0.29 | 1.0 | 0.48 | 0.18 | 0.19 | 0.15 | 0.11 | 0.055 | 0.0018 | 0.10 | 0.087 | 0.15 |
| pH | 4.3 | 4.5 | 4.7 | 4.6 | 4.4 | 4.9 | 4.9 | 5.1 | 5.2 | 4.8 | 4.8 | 4.8 | 4.8 | 5.9 |
| Rb | 0.79 | 1.3 | 1.9 | 0.71 | 2.8 | 1.2 | 1.0 | 1.7 | 2.5 | 1.9 | 0.68 | 0.83 | 0.82 | 1.8 |
| Se | 0.15 | 0.16 | 0.13 | 0.15 | 0.10 | 0.17 | 0.19 | 0.14 | 0.23 | 0.032 | 0.070 | 0.15 | 0.86 | 0.085 |
| Si | 8400 | 7900 | 8100 | 8000 | 6300 | 6800 | 6300 | 7400 | 7900 | 5100 | 4500 | 4400 | 4200 | 7900 |
| SO4 | 4900 | 5200 | 5200 | 5400 | 4400 | 4800 | 5200 | 5400 | 5700 | 3700 | 3700 | 4500 | 4200 | 8300 |
| Sr | 14 | 14 | 14 | 15 | 10 | 17 | 20 | 20 | 20 | 12 | 12 | 9.8 | 9.7 | 17 |
| Th | 0.27 | 0.20 | 0.18 | 0.17 | 0.13 | 0.067 | 0.055 | 0.068 | 0.084 | 0.0046 | n/a | n/a | n/a | n/a |
| Ti | 1.4 | 1.1 | 0.83 | 0.96 | 0.87 | 0.38 | 0.25 | 0.18 | 0.20 | 0.065 | 0.045 | 0.076 | 0.060 | 0.083 |
| Tl | 0.028 | 0.030 | 0.032 | 0.021 | 0.028 | 0.026 | 0.017 | 0.017 | 0.041 | 0.0091 | 0.0038 | 0.0091 | 0.0070 | 0.00094 |
| TOC | 33 | 24 | 19 | 21 | 26 | 11 | 8.5 | 6.6 | 5.4 | 5.2 | 2.8 | 1.7 | 2.0 | 1.6 |
| U | 0.13 | 0.093 | 0.11 | 0.079 | 0.084 | 0.062 | 0.043 | 0.092 | 0.19 | 0.014 | n/a | 0.00081 | 0.00050 | 0.0095 |
| V | 2.5 | 2.7 | 2.7 | 2.1 | 0.97 | 0.86 | 0.62 | 0.28 | 0.49 | 0.070 | 0.036 | 0.030 | 0.035 | 0.032 |
| Zn | 9.2 | 12 | 16 | 13 | 15 | 6.6 | 6.8 | 5.6 | 7.4 | 6.3 | 3.1 | 3.5 | 4.3 | 3.4 |
| Zr | 450 | 300 | 300 | 280 | 220 | 120 | 92 | 120 | 140 | 11 | 0.69 | 1.8 | n/a | 16 |





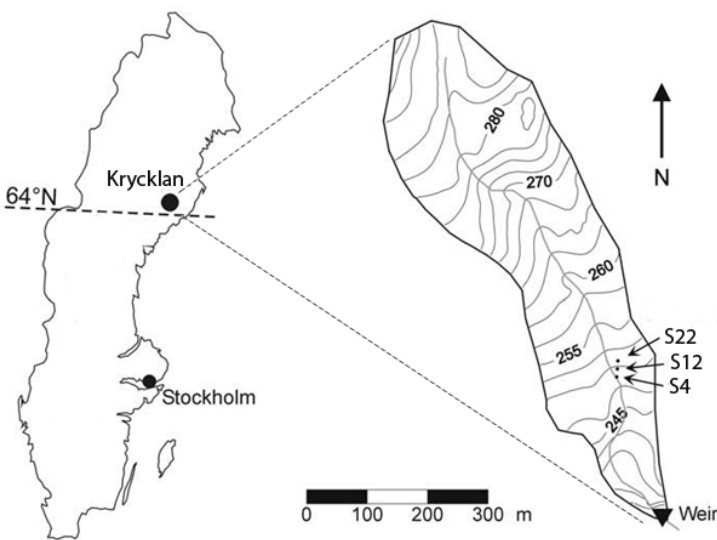

**Figure 1. Map of Sweden showing the location of the Krycklan catchment (left) and the Västrabäcken (C2) catchment (right). The location of the sampling sites (S4, S12 and S22) in the investigated transect are shown along with the elevation above sea level (m). Note that the lysimeters are located along the flow direction of the groundwater and not perpendicularly to the stream.**





**Figure 2. Average soil water and groundwater concentrations of TOC (top), vanadium (middle) and sodium (bottom) in the investigated transect. The area of the bubbles is proportional to the concentration, which in turn is given next to each bubble. The**



**background illustrates the LOI (%) in the two upper examples. The bottom subplot illustrates the approximate groundwater levels at different stream discharge. The distance from the stream is measured along the flow pathway of the groundwater.**

**Figure 3. PCA biplot for the average concentrations in each of the lysimeters. Most lysimeters located at the same distance from the stream ended up close to one another.**





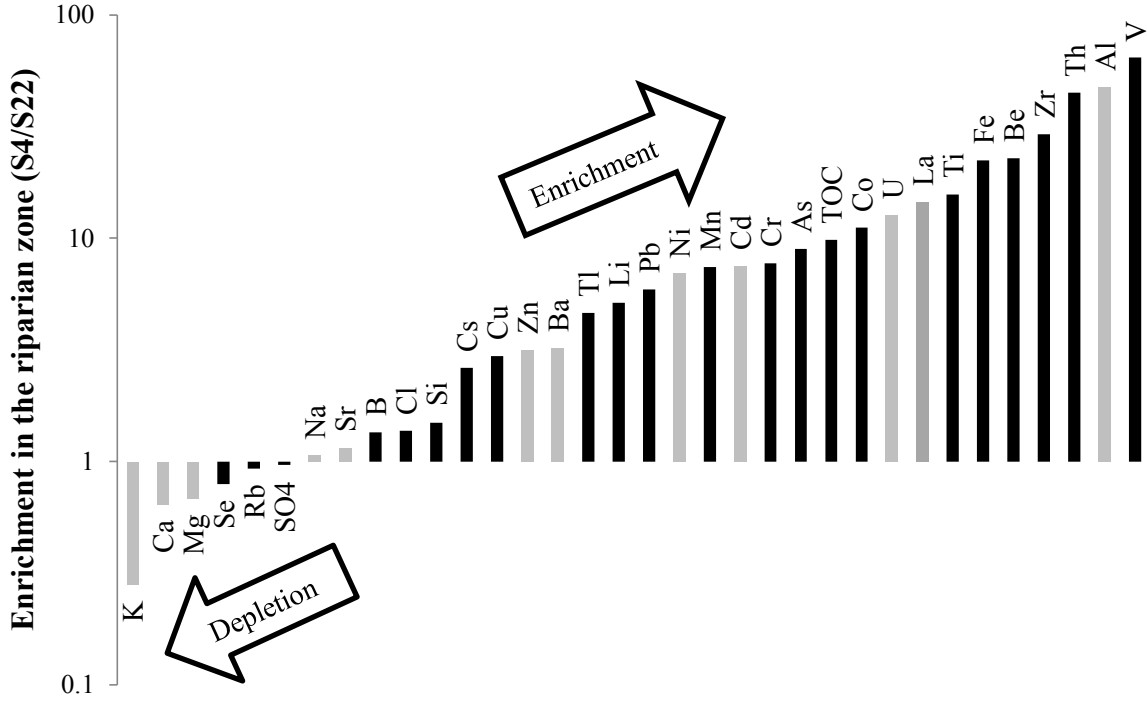

**Figure 4. Relative enrichment in the riparian soil water and groundwater expressed as the ration between the average concentration in the riparian profile (S4) and the average concentration of the upslope podzol profile (S22). Elements marked in grey are included in the regression in Figure 5.**

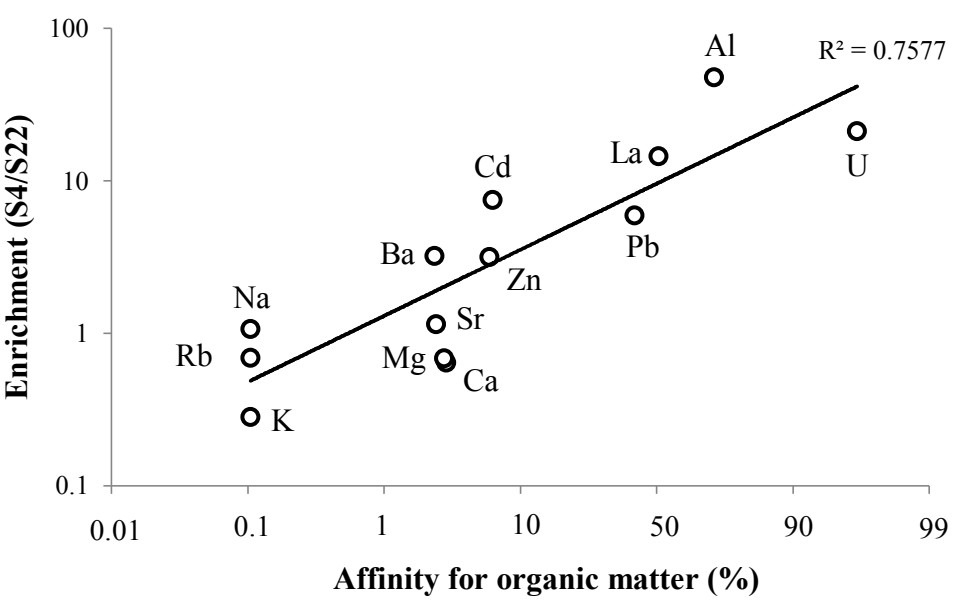

**Figure 5. Enrichment in average groundwater/soil water concentration when comparing the riparian zone (S4) and the upslope podzol (S22) as a function of the affinity for organic matter. The affinity for organic matter is plotted on a logit scale (Eq. 1).**





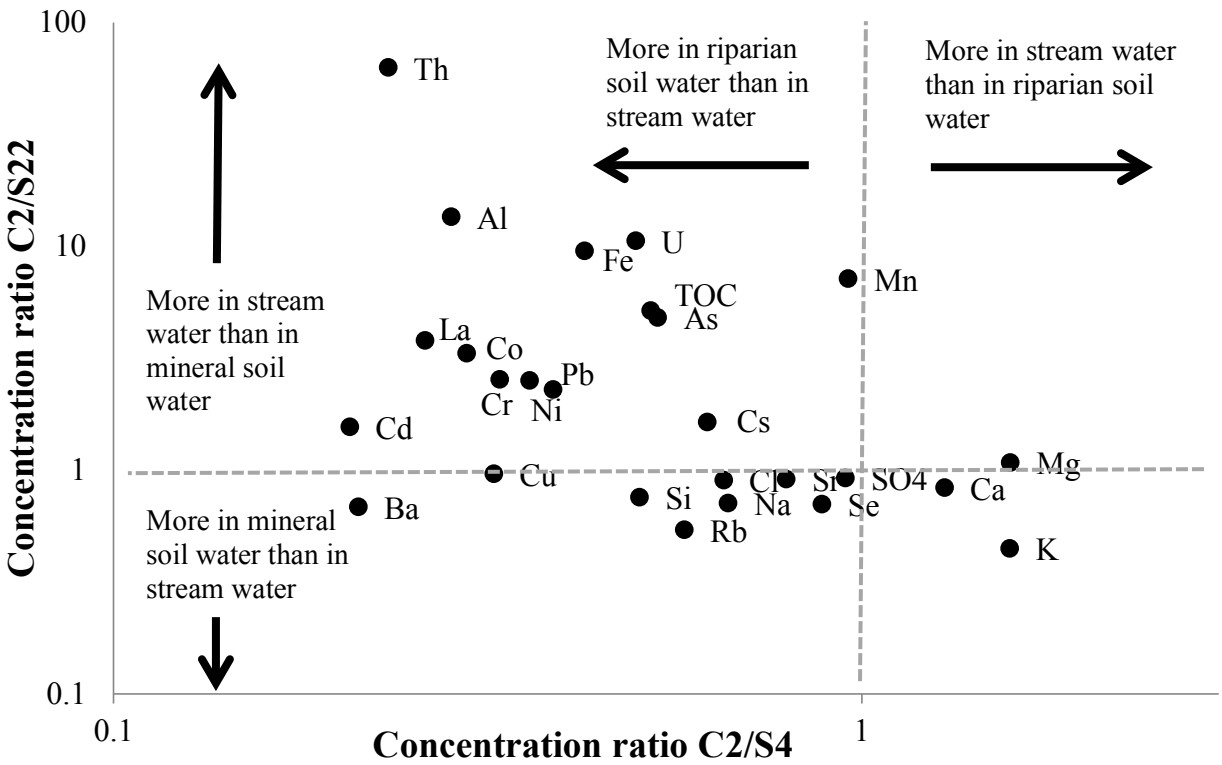

**Figure 6. Ratios between the average concentrations in stream water (C2) and riparian soil water (S4) (horizontal axis) and between the average concentrations in stream water (C2) and mineral soil water (S22) (vertical axis). Elements above the dashed horizontal line occur in higher concentrations in the stream water than in the mineral soil water. Elements to the left of the vertical lines occur in higher concentrations in the riparian soil water than in the stream water. The selection of elements is based on the available measurements in this study and the stream water data presented by Lidman et al. (2014).**





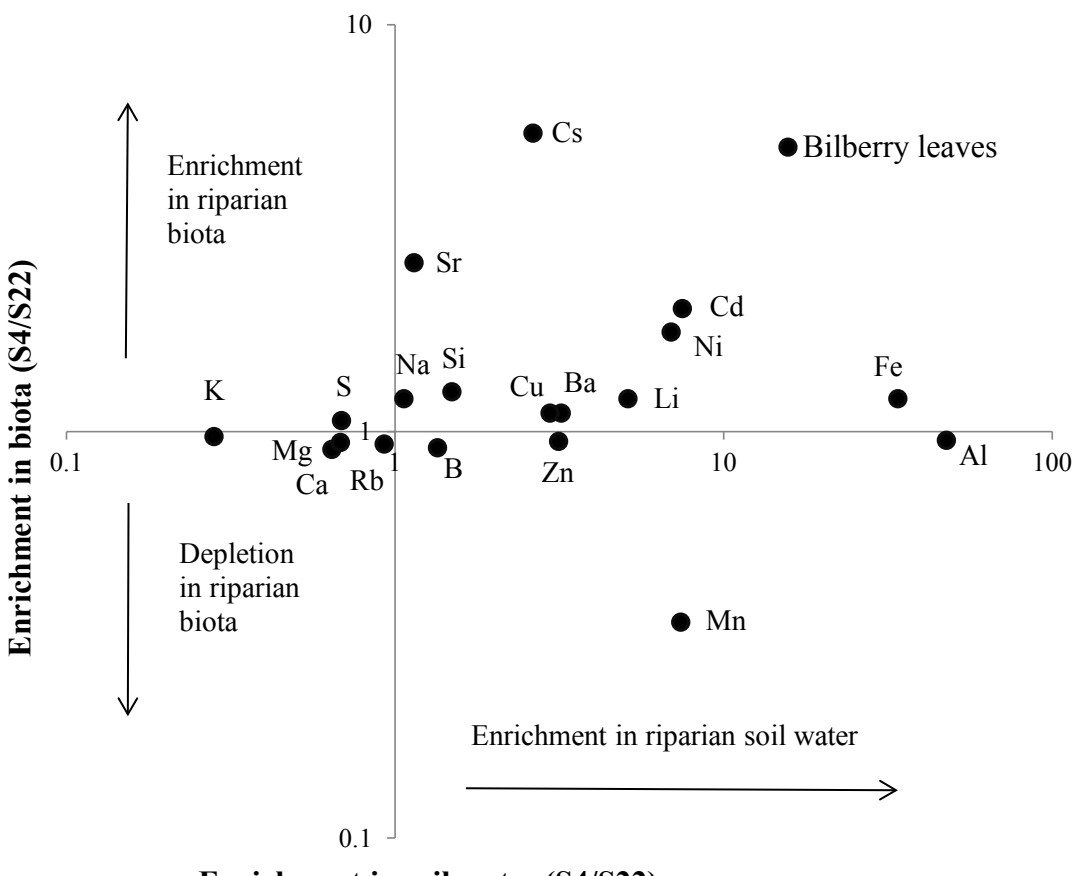

**Figure 7. Enrichment of elements in bilberry leaves vs. enrichment in soil water when comparing the uphill site (S22) and the riparian site (S4).**

