# Peer review of "From soil water to surface water – how the riparian zone controls the transport of major and trace elements from a boreal forest to a stream"

_Biogeosciences, 2016_

## Referee Comment (RC1) · Anonymous Referee #1 · 10 Jan 2017

General Comments:

I read the manuscript "From soil water to surface water – how the riparian zone controls the transport of major and trace elements from a boreal forest to a stream" by Fredrik Lidman and coworkers. Generally, I think that this manuscript quite good structured and provides new ideas/insights on the role of the riparian zone in chemical transport (affinity to organic carbon). Many of the graphs provide creative ways to explore the large dataset. This way, new insights into transport mechanisms related to the riparian zone are provided.

Generally, the abstract could highlight the main findings of this study more clearly. Additionally, the abstract could more clearly point out the "limits" of the state-of-the-art.

Currently, the start of the abstract reads more like an introduction that points out the aim, but somehow does not strongly motivates the innovation/importance of the current study. Altogether, such changes to the abstract would be beneficial for the article, as it could attract more potential readers.

In several cases, clarifying the text, shortening sections and being more "to the point" might help to convey the message (especially in cases where the same message is repeated several times in a row). Overall, I think the article is creative, innovative and would be very suitable for publication. Most of the comments are meant to improve the general understanding and readability of the paper. Please consider my comments as minor revisions to the manuscript.

Small overall comments:

- A s some of the graphs do not provide additional information, the number of graphs could be decreased.

- A shortened title for the manuscript would certainly be beneficial. The title could also be a more direct sentence regarding the key finding in the paper.

- I sometimes miss direct comparison to other papers with a large amount of elements, e.g. what new message does this bigger data set provide e.g. in comparison to Lidman et al., 2014? Perhaps this could be more clearly expressed in the discussion of the results.

Specific Comments:

Abstract:

1. The abstract could be improved by stating the current gaps/limitations in one sentence in the abstract, e.g. the reason for investigating the large group of trace and major elements.

2. One of the strongest messages in the manuscript is probably that the enrichment of

water in riparian zones is linked to the affinity with organic C, as the authors wrote in L. 20 – 22 (P1). I believe it would improve the manuscript if the results could stand out more clearly in the abstract.

3. The last two sentences of the current abstract are quite general (L. 24 – 26, P 1). In general, the fact that the riparian zone plays an important role in chemical transport is not new (Authors even cite this themselves in L. 3-5 (P2, Introduction)). It would be great if the implications of the study (summarized) could be closer related to the results, e.g. would focus more on the innovations. There are sufficient other interesting mechanism-ideas and results in the paper that do not really appear in the abstract.

Introduction:

4. In the first paragraph, the importance of the riparian zone is mentioned multiple times. This is okay, but sometimes the reasoning for different sections is relatively similar (e.g. the importance of the riparian zone on river water quality and lake water quality is mentioned in L. 3-5 (P2), L 11-13 (P2) and L13-15 (P2). I would advise to shorten this section of the introduction If the focus of these sentences was meant differently, and the information is crucial for the introduction, authors need to rewrite this to make the difference in "emphasis" of both sections clearer.

5. Although the second part of the first paragraph is more focused on the "state-of the-art", L. 23 -25 (P2), closes the section again with the conclusion that the riparian zone is important for stream water chemistry. As read from comment 4, it might be nicer to end this section with a conclusion related to the literature. Improvements as proposed in comment 4 and 5 could make the text more concise and to the point.

6. In L. 26 the authors state that the role of a large amount of elements remains unexplored in the riparian zone, after summarizing a large group of elements in L 13 – 20 (P2) that already have been studied. Although I understand that there are indeed many more elements that have not been explored, the importance of a large amount of these elements is not discussed and might be limited for this type of system (e.g.

raises the question: why would we care?). Here, I miss the link to the hydrological importance of other elements that have not been studied. This would also strengthen the statement afterwards that the riparian zone also plays an important role for other substances (this is important, because?).

7. Related to comment 6, the general reasoning that this study tries to obtain a more over-arching understanding of the riparian zone and the transport of elements is much stronger and could be more in the foreground as a reason to do this research. In that perspective, it would be useful to add relevant background-literature that already tries to understand and connect the transport of different elements (this is currently missing in this section).

Materials and methods

8. The authors describe the sampling locations in relation to the flow pathways of the groundwater. How are these flow pathways defined? What do these flow pathways look like? It would also be great to see the flow pathway along the 22 – 12 – 4 m transect. This would also be interesting related justification for using the stream water chemistry station 300 m. downstream.

9. At L. 25 (P3), there is a detailed description of the soil type of S22, but soil type information from the other two stations does not follow. It would be good to also mention this information for the other lysimeters. Now you have to somehow obtain this from L. 28-29, but this is not written directly. It would be great to have a profile description for S4, 12 and 22, so that you know in which soil layers the sampled water is located.

10. There is information about the depth of the organic layer, but the data is not 100% consistent, e.g. the exact depth of the organic layer for S22 is not directly given (e.g. L 26, is rather indirect with just giving a % below a certain depth). Additionally, I was wondering if there is a general soil map of the catchment. This is also related to the validity of the downstream water chemistry station. For example, Stockinger et al., (2014) (Seasonal soil moisture patterns: Controlling transit time distributions in a

forested headwater catchment, WRR) reported about differences in transit time for a 38.5 ha stream with sampling stations that have relatively small distances. The map does not give any information about the differences in width of the riparian zone – soil type distribution.

11. Table S1 (not 2 as mentioned in the text) shows the porosities of the soil for different depths. Generally it is not clear why (1) porosities for certain depths are given (2) certain depths in the lysimeters are sampled. Is this related to soil layers/ groundwater conditions? In 2-3 (P4) the authors mention that the concentrations at S4, 12 and 22 are typical and refer to Figure S2. Some questions arise when looking at this graph. First of all, at what depth is the organic content measured and what do the quantile values represent? Secondly, in relationship to this 20 m. "catena" approach, where would the concentrations at the lysimeters show up?

12. The second part of section 2.1 represents more general information about the region, whereas the first part presents more specific information about the lysimeters. Would it not be more logical to twist this around, e.g. more general information first, specific information about the lysimeters after that?

13. Related to the profiles, the description of the sampling positions, including the depths that are not/ more frequent saturated (highest groundwater), it would definitely help the paper to schematically show the profiles, the sampling locations, the soil layers (as mentioned before) and the groundwater levels during different flow regimes (as presented in the results).

Results and Discussion:

14. Regarding table 1 and S3: an ANOVA analysis could be useful to determine the differences in concentrations for the different depths are significantly different.

15. I am not sure if Figure S6 and S7 are really necessary. An addition to table 1 with information on the trend of all measured elements would be more useful. The results

as shown in S6 S7 could just be presented in written text, with a reference to the table.

16. L 15-17, P 8: As the authors are referring to Table 1, to state that certain elements have higher concentrations in the riparian zone. It would be great if Table 1 could somehow summarize which elements have higher/similar concentrations in the riparian zone, perhaps even with significance (significant difference, e.g. ANOVA). I like that Figure 4 summarizes this, but providing such information in Table 1 would definitely help (e.g. give the enrichment in the table).

17. Figure S4 presents interesting results, which might be important enough to bring to the main figures.

18. Section 3.6: the title of this section is a bit misleading. The section mainly discusses the more general implications of this study. It would be great if the title of this section could indicate this. One could additionally argue if part of the text here could be shortened – more to the point/ part of this section is more appropriate for the conclusion session.

General:

19. The manuscript contains several phrases with unprecise wording in it, such as "more or less", which makes statements in certain sentences sound less precise. An example can be found in the second sentence of the abstract: "In the boreal region the riparian zone of headwaters often tends to be -wetland-like with high concentrations of organic matter, low pH and more or less reducing conditions." Please consider rephrasing such sentences, especially in "fact-based" cases.

Technical Corrections: T1. Line 18-19 and 20-21 P3. Name of the measurement locations is clear and does not need to be justified (would be too much off track). Additionally, in L. 23 (P3), the station names are again introduced, but now with more detail. Information about the station names could just be directly added to the sentence in Line 18-19 (P3). Explanation on depth could come around L23, but does not need

to be explained in detail afterwards.

T2: Line 4, P5: change "draught" to drought.

T3: Figure S1 is introduced after S2, please reorder figures according to order of occurrence.

T4: Add Loss on ignition (LOI) to the caption – as far as I could trace back LOI was not given before. This also occurred in other cases (e.g. S6 and 7 mentioned before S4) and should be avoided.

T5: P6, L 27, rewrite last sentence.

T6: P7, L 3-4. From the rest of the text, I understood that there is one profile with different sampling depths that are sampled during 10 different events. Here, it reads as if at location 22, there are multiple profiles. Please consider rewriting.

---

## Referee Comment (RC2) · Anonymous Referee #2 · 7 Feb 2017

General comments

This manuscript follows the idea that riparian zones are mobilization and mediation zones for more than just DOC and nitrate as shown in numerous studies. For that purpose a very good dataset embedded in the Kryckland experimental catchment is utilized. The study shows that indeed a lot of elements are enriched in the riparian zone and partially translates into the stream. However, the way how the manuscript introduces the subject, raises key questions, explains methods and present and discuss result is not sufficient at the moment. On the one hand, the manuscript is repetitive in many parts, circling around the statement of the relevance of the riparian zone without adding much knowledge in the and. On the other hand the manuscript fails to bring

most things to the point and often lacks a clear structure within the chapters. Methods are not really reproducable and vague in terms of sampling and statistical analyses such as PCA. Own results, results of other studies and discussion are often mixed and hard to distinguish.

Overall, I expected more from this studies: After reading the manuscript I see the point that there are certain elements enriched in the soil/ groundwater of the riparian zone but are left alone with 1) the transfer to the stream and filtering/ mediating processes at the interface that explain differences to the stream water quality and 2) the question if the elements are really enriched in the riparian zone or just more mobile. I know that additional samples are hard to take and to include but modelling (kind of toy modelling as a showcase for dominant processes) would be a potential solution. Taken all this together I cannot support publishing the paper in the present state.

Below, specific comments can be found: Abstract

Page1 L9: I am not fully sure on the definition of riparian zones but I did not know that this includes lakes. Is there a reference for that later on in the text?

Page1 L10: "more or less" is a bit imprecise – please use a better phrase for that

Page1 L12ff: I expect from an abstract to state the problem and shortly describe where and how the study took place. This is not the case here and should be corrected.

Introduction:

Page2 L5ff: What is the difference between water qulity in the first sentence and "fluxes of many nutrients, pollutants and other substances" in the second? Isn't it the same?

Page3 L5f: The hypothesis should be sharpened: "it might be possible to identify some of the key processes" is not sufficient. What exactly, based on the literature review above, do you aim at?

Material and Methods:

The entire section on the study site is a wild mixture of different levels of information. I suggest to restructure this a bit, starting large from the general catchment description (location, climate, land use, hydrology, soils...) to the details of the specific transsect.

Fig. 1: This figure suggest that the show catchment actually is the Kryckland catchment. There is enough space to show the nested approach and location within Kryckland. Any other information such as extent of the riparian zone/ soil map would be helpful.

Page3 L18f: Just the transect does not tell you the groundwater flow direction. So, where is this information coming from? Topography? Other wells and head measurements?

Page3 L25f: "probably typical"? Are there references that you may cite for typical soils in this area?

Page3 L26: What is the meaning of "mor" in brackets?

Page3 L32f: I don't understand this sentence.

Page4 L2f: The accumulation of organic carbon is already stated in the introduction. F you want to be more specific on this particular site, give a reference for that.

Page4 L31ff: I have problems to imagine the sampling: You are talking about lysimeters? As far as I know, lysimeters are devices to measure soil water flux/ recharge/ ET quantity? So, how exactly are samples taken? With suction cups? Only in the saturated zone? Please make that clear, provide references or a figure!

Page4 L13: I don't understand this regression. What is the predictor and what the response? And why aren't there just measurements of the heads?

Page5 L23: Is this relationship established by you or published before? This is not clear here.

Page6 L 5: Can you summarize omega here and give the units?

Page6 L 6f: Please provide mor details on the PCA: Scaling of variables? Rotation? Kaiser critarion for selection of number of components? What data have been used? Mean concentrations or all snapshots in time?

Page 6 L 10ff: I would like to see an introduction to biological uptake in the introduction section. Are there studies which did that before? What about fractionation? Why did you do that at all? For a mass balance of elements or as an independent sample? Please elaborate on that a bit more. Results and discussion

In general I have problems separating the results from this study from previous published material. Moreover, description and discussion are often presented in a mixed form. I suggest to correct this.

Fig. 2: The bottom subplot doesn't show groundwater levels as noted in the caption but hydraulic conductivity. Please correct that.

Table 1: Exhausting information but I see the point showing this data. However, to better capture information I suggest to indicate horizontal (between the profiles) enrichtment/ dilution/ chemostasis by additional column and upward, downward, sideward arrows or something like that. What about the temporal scale? Are there changes of concentration over time and where is that shown? You should somehow justify that the mean is a suitable measure of concentration over the year.

Page7 L3f: What do you mean by "stable pH in all profiles of S22"? S22 is one profile with different depths, right?

P7 L8ff: This longer section on Fe speciation is based on other studies only and not on own results? This would make it more suitable for the introduction or a later discussion but not for the result section!

Page7 L 19ff: I cannot find the details on factor loadings and explained variance. You should provide that. Are all nine components fit the Kaiser criterion?

Page 7 L25ff: I would like to see a description of the general results, different factors,

explainability of different elements instead of directly coming to the deviations and exemptions. In general I miss the outcome and meaning of the performed PCA. PCA is not just a look at the biplot!

Page 13 L1ff: The chapter on the importance of the riparian zone circles around the one key finding of this study: Most elements can be found in higher concentrations in the riparian zone as a function of organic carbon. There is a lot of redundancy with introduction and other part of the result chaper: e.g. the often stated fundamental role of riparian soils for discharge generation and substance mobilization. These parts should be shortened here and the focus shifted to the reason of high element concentrations, weathering, transport...

I tend to say that the inclusion of solid phase samples would make the manuscript much stronger: Are higher concentrations just a function of mobilization in presence of TOC or locally different weathering rate?

---

## Referee Comment (RC3) · Anonymous Referee #3 · 10 Feb 2017

This is an interesting and generally well-written paper presenting the effects of the riparian zone on the transport of elements – in fact how organic matter is affecting the dynamic of different elements. This paper also documents the great value of long-term monitoring data from catchments having a well-thought sampling design. I particularly like the figures illustrating the effect of the riparian zone. However, before the manuscript might be published in Biogeosciences, it needs some revision (moderate), particularly a shortening of the whole manuscript. There are some repetitions which can easily be removed. I did not count how often the authors mentioned that higher concentrations of most of the elements occurred in the riparian zone. It is a pity that the pH as a very crucial parameter was not measured. I would like to see a short

explanation about the potential estimation error of the pH on the results and I would not use the pH for further statistical analysis as the PCA. Furthermore, I would shorten the description of the results regarding pH values (keep the limitations in mind). I do not agree with the approach to estimate biological uptake. The authors just collected some leaves and measured concentrations of some elements in these leaves. This is not uptake – just a concentration. Please consider that in your manuscript and add a short description about the meaning of these element concentrations. The whole section related to biological uptake can be largely shortened. In section 3.6 you are referring to a change from relatively dry organic-poor mineral soils to wetter organic soils. Do you have any quantitative information about this gradient, i.e. contents and / or stocks of soil organic carbon?

Some minor comments: There is no "organic content" (e.g. page 3, line 27). Please be more specific (e.g. content of organic carbon). Please explain acronyms (e.g. TOC: page 3, line 5; REEs: page 5, line 18). Figure 2: I would indicate the thickness of the organic soil horizons in the upper panel.

―――――――――――――――――

---

## Author Comment (AC1) · 9 Mar 2017

We thank all the Referees for their insightful comments on our manuscript. Our impression is that all Referees found the dataset interesting and worthy of publication, but there were also many comments concerning the presentation and the structure of the manuscript. Having carefully considered all comments, we believe that most concerns could be addressed by slightly revising parts of the manuscript. Below we respond to each of the remarks made by the three Referees:

Comments to Referee #1:

General comments: Referee #1 states that the manuscript is "quite good structured

and provides new ideas/insights on the role of the riparian zone". He/she also thinks that large dataset has been explored in creative ways so that new insights into the functioning of riparian zones are provided. However, Referee #1 also thinks the abstract should be rewritten so that the main findings of the study are better highlighted and the motivation and limitations of the study are clearer. We believe that these are valid remarks, which call for slight revision of the abstract.

Overall, Referee #1 also thinks that clarifications and shortening of sections throughout the manuscript would improve the presentation. Having read our manuscript anew, we feel inclined to agree with Referee #1 that there are parts of the manuscript, which should be shortened and/or clarified.

Small overall comments:

Referee #1 thinks that all graphs are not necessary. It would be helpful to know which ones should be omitted in his/her opinion, but if we are given the opportunity to revise the manuscript, we will try to consider how the number of graphs could be decreased. There is always the option of providing additional graphs as supplementary material.

The title of the manuscript is admittedly quite long, and we agree that a shorter title would be good. We should certainly give this some more thought, but at the same time it is important advertise what can be found in the manuscript.

Referee #1 misses a direct comparison between our results and other papers with a large number of elements. The main reason is that we have not encountered any similar datasets, which could be used for such comparisons. Referee #1 specifically mentions a paper by Lidman et al. (2014), which also was cited in the manuscript. A major difference is, however, that that paper deals with stream water, whereas this manuscript focuses on soil water. We must also emphasize that we used the data published by Lidman et al. (2014) to compare the element concentrations in the uphill soil and the riparian zone to the concentrations in the stream. This can be seen in Fig. 6. It is not obvious to us how further comparisons between these two datasets could

be made.

Specific comments:

1. We agree that the reason for investigating the large group of elements in the riparian zone should be stated in the abstract.

2. We also agree that the main findings of the study should be clearly stated in the abstract. The importance of the affinity for organic C certainly belongs there. This should be emphasized in the abstract.

3. Referee #1 remarks that the final sentences in the abstract are too general. We think that this is a valid observation so we agree that we could be more specific here.

4. Having read the introduction again, we acknowledge that there is a degree of repetitiveness in the introduction. It should be possible to shorten this part a bit, as suggested by Referee #1.

5. Again, we agree that the introduction could be written more concisely.

6. The manuscript includes a large number of elements, which previously have not been investigated in this type of environments. The motivation of the study could be further emphasized by explaining why each if these elements is interesting, as Referee #1 suggests, but at the same time the number of elements is too high to allow a thorough discussion of all of them. In our opinion there are two important reasons to look at all these elements. First, all of them are of interesting in their own right, e.g. as micronutrients, pollutants etc. There are environmental issues related to essentially all elements in the periodic table and the transfer between terrestrial and aquatic systems is often overlooked. Second, including a large number of elements allows the comparison of elements with similar or contrasting biogeochemical properties, which can help to distinguish what processes are important for the fate of various substances in riparian soils. This is for instance illustrated by Fig. 5, where elements with different affinity for organic matter are shown to behave differently in the riparian zone. We

agree, however, that we should try to explain this better in the manuscript.

7. We agree that the objective of the study is to achieve a more over-arching understanding of the riparian zone and its role for different types of elements. As commented above, we might need to put more emphasis on this. We could also try to add more references to previous studies, but a challenge is to find papers, which deal with element transport on a more general level. In our experience, most papers tend to focus on one or a few elements, and we do not think that it is possible to cover all of the included elements in detail.

8. The flow pathways in the transect were identified in tracer experiments before the lysimeters were installed in 1995. According to previous studies (cited in the manuscript), the flow pathways are strongly dependent on the hydrological conditions, in particular the groundwater levels. This is illustrated in the Fig. 2 (bottom). Perhaps it would also be helpful to add a photograph of the transect in the supplementary material.

The reason for using stream chemistry sampling 300 m downstream of the transect is that this is the closest sampling location in the Krycklan catchment. Note that it is a first-order stream that does not receive water from any other landscape type. We must also emphasize that even if the stream chemistry would have been sampled right next to the transect, the would still not guarantee that the transect would have been more representative for the catchment of the stream at that location. On the contrary, the current setup with the stream stretching both upstream and downstream from the transect might even make it more representative for the catchment as whole, since the stream gets increasingly larger further downstream.

9. The reason that S22 was described in more detail than the other two soil profiles was that S22 unlike the others has distinct soil horizons. We agree, however, that we could elaborate more on how the organic content of S12 and S4 varies. Note that this also is presented in Fig. S1.

10. There are soil data from the Swedish Geological Survey (SGU) for the area, but unfortunately it does not provide much insight into what the soils really look like. The catchment is dominated by till (100% according to the maps) because the organic soils of the riparian zone are not mapped. One reason is certainly that the importance of riparian zone historically has not been recognized, and this also makes it hard to assess the importance of organic-rich riparian soils on larger scales. We have investigated the soils along this stream and know that the accumulation of organic matter near the stream occurs all along the stream channel. This is illustrated in Fig. S2. However, we do not think that it is a good idea to add this information to the map because the riparian zone is too narrow in comparison to the catchment as a whole. The width of these organic soils is likely to vary depending on the local typography and the hydrology, and attempts to model this are being made. We hope that this will lead to a better description of the extension of organic riparian soils.

11. The reference to Table S1 is clearly incorrect, as Referee #1 has commented. We agree that it may not be crucial for the manuscript to show that soil porosity at different depths. We added this information to the supplementary material just to make it accessible and citable. The depth of these samples as well as the depth of the lysimeters are based on the sampling and the installation of the lysimeters in 1995. As regards Fig. S2, we need to add some more information concerning how the data was collected and describe more thoroughly what it illustrates. One alternative might also be to add the investigated transect to the figure.

12. It might be more logical to present general information about the area before presenting the transect, as suggested by Referee #1. Since the transect is the focus of the manuscript, we thought it would be easier to start by describing the lysimeters etc., but this could easily be changed.

13. In order to limit the number of figure in the manuscript, we tried to include most the requested information in the subplots of Fig. 2. We could prepare a separate illustration of the transect, if it does not suffice with a reference to Fig. 2.

14. We agree that ANOVA could be used to further analyze the data statistically, but the question is whether it is motived to include that in this manuscript. We feel that we would not be able to go through all the results for all the elements. Perhaps this is something that can be kept in mind for future studies that focus more on specific elements?

15. We agree that Figs. S6 and S7 are not necessary. They were included because we thought that some people might be interested in those elements, but the figures could certainly be excluded without loss of context. We will not insist on keeping them if they are perceived as unnecessary.

16. We think that it would be a good idea to add more information to Table 1 so that it becomes clearer how different elements behave. The only limitation is that the table already is quite large, but we could certainly see if it is possible to find space to add the information that Referee #1 requests.

17. We agree that Fig. S4 is interesting. It would certainly be an option to transfer it to the main figures, as Referee #1 suggests.

18. Referee #1 thinks that the title of section 3.6 is a bit misleading and that parts of the text could be shortened or moved to section 4. We agree that section 3.6 should be renamed and slightly revised.

19. "More or less" is not a good wording. We should remove such unprecise descriptions from the manuscript.

In addition to these comments, Referee #1 has a number of technical remarks, which we agree need to be addressed.

Comments to Referee #2:

Referee #2 is arguably the most critical of the three referees, expressing a certain disappointment with the results of our study. Nevertheless, it is acknowledged that the dataset is "very good", but Referee #2 has concerns regarding the presentation,

both because the text is considered too repetitive and because it is not sufficiently to the point. In this respect, Referee #2 largely echoes the comments of Referee #1. As authors, it is hard to argue with such statements so we evidently need to think about how to present the results and structure the manuscript to make it more readable. Having read our manuscript anew, we agree that there is a certain degree of repetitiveness that should be addressed.

We do not fully understand what Referee #2 means when he/she states that the methods "are not really reproducible". The results are obviously not reproducible in the sense that we cannot collect samples under exactly the same conditions as when these samples were collected, but this is of course the case for nearly all environmental studies. The suction lysimeters are, however, still left in the soils and would be possible to sample again. Indeed, they are still being used for other studies. More importantly, however, it would be possible to install similar equipment at other sites and do corresponding measurements. Whether that would reproduce the results of this study or not can obviously be known in advance, but it would certainly be possible to design and conduct a similar study. As for the description of the sampling and the statistical analyses, however, we arguably need to be more specific, as Referee #2 thinks that they we are too vague. This is certainly something we could improve.

Referee #2 states that he/she expected more from this study, particularly raising two questions that he/she evidently thinks we should have answered. The first question concerns the difference in water quality between the riparian zone and the stream. Perhaps there is a need to collect samples even closer to the stream, but with the current data we can only compare the water chemistry 4 m from the stream to the water chemistry in the stream. It is possible, as Referee #2 discusses, that there are filtering/mediating processes that we fail to see with the current experimental setup, but it is also possible that the differences are caused by variations along the stream. One must keep in mind that we only sampled a single transect along ca. 1 km long stream. As shown in Fig. S2 and discussed in the manuscript, there are longitudinal variation

in the composition of the riparian zone, which means the investigated transect may not be entirely representative for the catchment as a whole. Furthermore, there may be scale-dependent differences along the stream even at these scales, e.g. more inflow of older and deeper groundwater further down in the system. By the transect we have identified a compact till layer at ca. 1 m, which seems to limit the exchange with deeper groundwater, but this may not be the case throughout the entire catchment. Therefore, the lack of an exact agreement between the water quality of the stream and that of one single riparian profile should not raise too much concern in our opinion. Nevertheless, we believe that the gradient from uphill podzol soils to organic soils in the riparian zone describes something important about the functioning of boreal headwater catchments. We do not see how it would be possible to settle this question based on the dataset we present, but hopefully the observations of this study will lead to more research on the role of riparian soils, which eventually can shed more light on these issues.

The second question concerns the functioning of the riparian zone and its long-term mass-balance. Again, Referee #2 raises a highly relevant question, but we do not see how it would be possible to resolve that question based only on soil water data. As suggested by Referee #2, modelling and sampling of soil would be possible strategies to gain more insight into this matter. Indeed, the are modellers now working with this data, trying to integrate a hydrological transport model with a thermodynamic chemical model. We are also working with soil samples that could reveal more about what is going on in the riparian zone. It is not included in the manuscript, however, partly because we have not finished the analyses, partly because it would result in a very lengthy manuscript. Even in its current state, this manuscript is fairly long, and then it must also be acknowledged that we do not discuss any of the elements in detail. Nor do say much about the temporal variability in the soil water chemistry, although such information can be found in the dataset. Therefore, we do not think that it is a good idea to add even more data. It is already challenging enough to generalize the current results into a coherent manuscript.

Overall, however, we think that these two questions are highly relevant, and we hope that the research in the Krycklan catchment eventually can help to solve them. We can assure Referee #2 that this manuscript in no way was intended to be the final word on this matter. As mentioned, there are currently multiple on-going efforts to model the hydrology and the biogeochemistry of this particular transect. One reason why we think that is important to publish these results is that they should become available and citable for on-going and future research. Getting all the mass-balances right – both in the present and historically – is, however, not an easy task. It requires more data and more work, and therefore goes beyond the scope of this manuscript in our opinion. Therefore, we believe that this study must be seen as part of a greater effort to understand the long-term functioning of riparian soils and their impact on stream water quality.

Specific comments:

Page 1, line 9: Referee #2 may be right that the concept of riparian zones is limited to streams and rivers. This needs to be clarified in the manuscript.

Page 1, line 10: We agree. This should be rephrased.

Page 1, line 21: We agree that information on the sampling and the sampling location should be added to the abstract and that the problem should be clearly stated. Referee #1 also called for a revision of the abstract.

Page 2, line 5: Strictly speaking, we do not think that "water quality" is exactly the same as "fluxes" of various substances. Water quality is certainly related to fluxes, but when discussing fluxes it relates more broadly to the fate of various substances in the environment and not only in the stream water. However, it may be superfluous to bring up both terms in two consecutive sentences so it is probably a good idea to rephrase this.

Page 3, line 5: Referee #2 observes that the hypothesis is a bit vague. We agree that

we should try to describe the objective and hypotheses more clearly. Site description: The site description indeed contains large amounts of information at different levels. We believe that it is important to provide an extensive background both the area and the specific transect with references to publications where more information can be obtained. In the current version the focus is on the transect, which was investigated, but it would be possible start with background description of the area and then zoom in on the transect, as the Referee suggests.

Fig. 1: The investigated transect is indeed located in the Krycklan catchment, which also is clearly stated in the manuscript. Fig. 1 does not depict the entire Krycklan catchment, but one of the small headwater subcatchments (C2 or Västrabäcken) within the Krycklan catchment, which is where this particular transect is located. As can be seen in Fig. 1, this is an appropriate scale for showing the location of the transect. When using a map of the entire Krycklan catchment, it is hard to even discern the subcatchment (C2), in which the transect is located, let alone the transect itself. Since no data from other subcatchments in Krycklan are discussed in this manuscript, we do not think that a map of the entire Krycklan catchment is strictly necessary in the manuscript itself, but it could be added to the supplementary material for those who are vaguely familiar with the Krycklan Catchment Study and wish to know where the S transect is located. All soils in the catchment (C2) are classified as till in the soil maps so it is not possible to them to map the extent of the riparian zone.

Page 3, line 18: Before the lysimeters were installed in 1995 tracers and a number of groundwater wells were used to determine the flow direction of the groundwater. In reality the direction of the groundwater flow probably varies somewhat over time depending on the groundwater levels, but that is hard to capture in this type of experimental setups that require permanent installation of sampling equipment. The direction also agrees well with what could be expected from the local topography, but that was not decisive when installing the transect.

Page 3, line 25: We agree that it would be good with better references here. To start

with we should include a reference to Fig. S2, which shows that the gradient from inorganic to organic soils is typical for the investigated catchment. There are similar measurements from other catchments in Krycklan, which show that small streams typically have highly organic riparian zones. We will have to check whether that information is citable somewhere or whether we somehow could include it in this manuscript. On larger scales it is quite hard to show that this type of organic riparian soils is common because their importance has not historically been recognized. For example, in soil maps from the Swedish Geological Survey the investigated transect is classified as till – despite its high content of organic matter. Riparian soils have, however, been investigated throughout the Krycklan catchment, and the results suggest that accumulation of organic matter is typical for riparian soils along the smaller (roughly first- and second-order) streams.

Page 3, line 26: Mor is a type of humus, which is typical for coniferous forests. It is not essential information so perhaps it should be removed in order to avoid confusion.

Page 3, line 32: This was an attempt to describe the area upstream of S22, but it evidently needs to be clarified.

Page 4, line 2: We do not entirely understand this comment. A reference is given to Fig. S2, which shows that accumulation of organic matter in the riparian zone is common throughout the catchment where this transect is located and, more specifically, that the LOI at the investigated site is typical for the riparian zone of this catchment. We do not see what other reference we would need to support our claim.

Page 4, line 31: We apparently need to describe the sampling better. There are different types of lysimeters, which can be used to measure different things. These lysimeters are best described as suction lysimeters and consist of small porous ceramic cups, which are buried at different depths in the soil profiles and attached to tubes. When connected to vacuum bottles water is extracted from the soils regardless of whether the cups at that time are in located in the saturated or unsaturated zone. Perhaps an

additional figure of the transect in the supplementary material would be helpful.

Page 5 (?), line 13: Normally there are loggers in groundwater tubes next to the lysimeters, which continuously measure the groundwater levels, but during sampling period the loggers were unfortunately out of order. Therefore, there are no direct measurements of the heads. Instead the groundwater levels (response) were estimated from the discharge in the stream (predictor), which is measured and logged at a weir downstream of the transect (Fig. 1). The full procedure is given in the cited reference, but it could perhaps be clarified also in the manuscript.

Page 5, line 23: The relationship was established based on older measurement because we unfortunately missed to measure pH in this sampling campaign. We should clarify this in the manuscript.

Page 6, line 5: Omega was originally defined in the cited manuscript (Lidman et al., 2014) as an index for how strongly different elements bind to organic matter. It is calculated by modelling the aqueous speciation and equals the fraction of a certain element that is predicted to be bound to DOC. Therefore, the unit is percent, although it can be thought of simply as an index ranging from 0-100, describing how strongly an element is expected to bind to organic matter in a certain environment. This should probably be explained in more detail in the manuscript, although a full description and discussion of this parameter is available in the cited reference.

Page 6, line 6: We agree that we should be clearer concerning the PCA (if it indeed should remain part of the manuscript). The variables were scaled because the concentrations of the included elements vary over several orders of magnitude. We used the average concentrations because that allowed us to include more elements in the PCA – otherwise there would have been more gaps in the data. As stated, Th and Cr were excluded because of missing values, whereas interpolation was used to fill in gaps for U and Zr. In addition, we failed to mention that pH also was excluded, since it was just estimated from other parameters. Since the PCA mainly was used to construct the

biplot (Fig. 3), no Kaiser criterion or other selection criteria was used. Likewise, no rotation was used.

Page 6, line 10: Once it had been observed that certain elements occurred in so much higher concentrations in the riparian zone than in the normal podzol soil, the question was raised whether the riparian zone could be an overlooked exposure pathway for various toxic substances. If the plants are exposed to nearly 100 times higher concentrations of certain elements in the riparian zone, this could lead to much higher concentrations also in, for example, bilberries, which frequently are being picked. This is the reason we made a screening of spruce shoots and bilberry leaves in the riparian zone. The results showed no clear increase in the element concentration in the riparian vegetation, suggesting the elevated element concentrations are present in forms with low bioavailability. We could certainly elaborate a bit more on this introduce the question better in the introduction.

One of the advantages of conducting a study like this in a well-investigated area is that there is plenty of background information available. It is important to put the current study in the context of previous research in the area. It is not always easy to know when it is best to introduce this information, but in some cases we believe that it is best to bring it up in the context where it is most relevant for the interpretation of the data rather than in the introduction or in the site description. One drawback is obviously that it is hard to tell what is based on previous publications and what is based on the present results. We have tried to be careful to cite previous research whenever references are made to older information, but it must also be acknowledged that some of the discussions and conclusions rely both on previous and present results. It should be possible, however, to make a stricter division between previous and present results, as suggested by Referee #2.

Table 1: We agree that the information is exhausting, but nevertheless those are the actual results of this study so it seems important include the table in the manuscript. The general trend is illustrated in

Fig. 4, but we could also see if it is possible to include the enrichment factor in Table 1 (without having to compress it too much). This was also brought up by Referee #1. The temporal trends are not shown in the table (or elsewhere) and are not discussed in any depth in the manuscript. The reason is that we do not know how we possibly could concentrate all that information into something that can be presented in a single manuscript. For some elements there is certainly considerable temporal variability, for others not so much, but we have not been able to discern any simple trend that could summarized in a similar manner as Fig. 5. It is very hard to come up with that type synthesis. Whether or not the average concentration is the most suitable measure over the year is again a very complex question, which ultimately depends on what issues one is trying to address. However, when trying to summarize such large datasets it is necessary to somehow make simplifications. There may very well be interesting patterns that we miss by using averages, but in any case we feel confident that we do not misrepresent the general patterns in the water chemistry by using averages. The idea is to make the dataset available to allow further analyses of, for instance, temporal variations.

Fig. 2: The bottom subplot does indeed show the approximate groundwater levels at different discharge as stated, but we should probably clarify that this is based on the data published by Laudon et al. (2004).

Page 7, line 3: Yes, it should be "at all depths in S22".

Page 7, line 8: The speciation of Fe is crucial for the mobility of many trace elements because Fe colloids have been found to be one of the most important vectors in boreal waters. The other important vector is organic colloids/TOC. The apparent absence of Fe colloids in the riparian zone is therefore important for the interpretation of the data, since it leaves TOC as the only plausible candidate for trace element transport. This is the reason that we wish to bring up the discussion about Fe at an early stage, which formally should not be a problem because the manuscript does not have strictly separated sections for results and discussion. We are, however, prepared to shorten

this section or move parts of it to the later parts of the manuscript. This would probably be good in some respects, as suggested by Referee #2, but we also think that it would be good to present all evidence that speak against the presence of Fe colloids before we go on and discuss the role of TOC. This is a quite lengthy discussion, which relies on the assumption that there are no Fe colloids present.

Page 7, line 19: The explained variance was only given for the first two principal components, which are the ones that are used in the biplot. The Kaiser criterion was not used to select which principle components to retain, since no further analyses were made using these results. We mentioned the fact that a large number of principle components were needed to explain most the variance just to illustrate that the problem could not easily be reduced to just a few dimensions, but this was perhaps not necessary. Likewise we did not provide or discuss any factor loadings, but all of this could of course be added to the supplementary material. We suspect, however, that anyone who has any interest in such things would be more interested in the full dataset, which we intend to make available. Then it would be possible to proceed with similar analyses for anyone who wishes to do so.

Page 7, line 25: The PCA was done at an early stage of the data analysis, and it is a part of the manuscript that we were uncertain whether to include or not. Referee #2 remarks that we focus too much on the biplot, which probably is why we though that the biplot was the most interesting result of the PCA. We particularly like the way the lysimeters group themselves into the three investigated soil profiles (S4, S12, S22), which is nicely illustrated by the biplot. This provides a good statistical justification for the proceeding analyses of the soil water chemistry, which largely relies on a grouping of the samples according based on the three profiles. To us this was the major reason why we wanted to include the PCA. However, this is no excuse for not providing a full description of the outcome of the PCA. Referee #2 is probably right when stating that we focus to much on the deviations and too little on the general results.

Page 13, line 1: Having read the manuscript anew, we feel inclined to agree with the

opinion of Referee #2 that there is a certain degree of repetitiveness and redundancy in the manuscript. It should be possible to shorten these parts. Referee #2 also thinks that we should focus more on the reasons of high element concentrations in the riparian zone – "weathering, transport ..." – which is a fair viewpoint, but the problem is that we do not yet have the full picture. In this manuscript we believe that we have demonstrated that TOC plays a key role, but it remains unknown whether the high concentrations ultimately are caused by increased weathering rates in the riparian zone or historical transport to from uphill soils to the riparian zone.

Referee #2 remarks that the inclusion of solid phase samples would make the manuscript much stronger. We certainly understand this viewpoint. In order to fully understand the functioning of the riparian zone it is not sufficient to just look at the soil water chemistry, as we have done in this study. The problem, as we see it, is the length of the manuscript. Already in its present state, Referees are complaining that the manuscript is too long. Although this partly may be related to some repetitiveness in the presentation, it must also be taken into account that the manuscript presents a dataset of considerable size. Most of the included elements are hardly discussed at all and are only briefly shown in tables and figures, and the temporal trends in the soil water chemistry are also largely overlooked, as discussed above. In our opinion there is a lot more that needs to be discussed in more detail in the data we present in this manuscript, and we think that it is likely that more publications will be based on this dataset in the future. The purpose of this manuscript was therefore mainly to present the dataset and try to describe and explain the major trends, thereby making it available and citable. Therefore, we do not think it would be wise to expand the manuscript further at this state by including even more data. However, in the light of these soil water samples we agree that it would be a good idea to also analyze solid samples from the transect. Indeed, such analyses being made, and we hope that the results soon can be published as well. It will hopefully help to answer some of the key questions that Referee #2 raises.

To summarize, this study was not the first step in trying to understand the riparian zone, and we must emphasize that it is in no way meant to be the last. As mentioned, analyses of the solid phase have been made and are currently being interpreted. We hope that it will be possible to answer some of the questions concerning the functioning of the riparian zone raised in this manuscript using these new results. Particularly, we hope that it will be possible use this data to discern weathering and accumulation patterns in the transect, which might answer the central question raised by Referee #2: "are higher concentrations just a function of mobilization in presence of TOC or locally different weathering rates?" Currently, however, we do not know, but obviously important processes are going on in the riparian zone, as illustrated by the results of this study. That alone we believe justifies their publication.

Comments to Referee #3:

Our impression is that the comments from Referee #3 overall are positive. He/she describes manuscript as "interesting and generally well-written", but also points out that the manuscript is a bit long, partly due to some repetitions that ought to be removed. In this respect, Referee #3 seems to agree with the other two Referees.

Referee #3 brings up the pH, which arguably is a very important parameter. We also regret that pH was not measured in these samples. It was certainly our intent to do so, but due a misunderstanding between the involved laboratories nobody actually did the measurements. When the mistake eventually was discovered, it was considered to be too late to make reliable pH measurements. Luckily there were pH measurements from other sampling campaigns in the transect, which enabled us to establish a relationship between pH on one hand and TOC and Ca on the other. The underlying assumption of this model was that pH mainly was driven by organic acids (TOC) and weathering (in this case represented by Ca). Thus, it was possible to reconstruct the pH with an average prediction error of 0.24 pH units, as stated in the manuscript. The relationship is also illustrated in Fig. S4. Referee #3 calls for an additional explanation about the potential prediction error, and it would of course be possible elaborate further on this

subject in a revised manuscript.

We agree that the prediction error in the pH might cause problems if it were included in further statistical analyses such as the PCA. Therefore, it was not our intent to include pH in the PCA, and as far as we can see pH is consequently absent in the biplot (Fig. 3). The statistics related to the PCA also do not include pH, and should we be mistaken here we agree that it should be corrected. However, we have not stated in the text that pH was not included in the PCA, and that is something that needs to be clarified in the manuscript. Furthermore, pH was included in the discussion about the PCA, although it was not part of the PCA, which is misleading. This should be corrected.

Referee #3 also criticized our approach to look that biological uptake in the riparian zone. We acknowledge that Referee #3 makes a valid point here. What we measured was indeed the concentration of various elements in biota, which is not the same as the uptake in biota. For instance, high uptake may not necessarily lead to high concentrations, since one also needs to consider the release. Therefore, this section has to be rephrased and, as suggested by Referee #3, shortened.

The reason that the element concentrations in biota were measured was that the high concentrations of certain elements in the riparian soil water potentially could lead to higher concentration also in the riparian biota. That would imply that the riparian zone could be an important exposure pathway for animals and humans with respect to many toxic substances. The crucial parameter in this case would be the concentrations in the vegetation rather than the actual uptake. This also brought up by Referee #2 so it is clear that we need to explain these measurements better in the manuscript.

Concerning the organic content of the soils the LOI (loss on ignition) for the three soil profiles is shown in Fig. S1. The variation in organic matter along the stream is further illustrated in Fig. S2. Referee #3 remarks that we need to be more specific when discussing for example the content of organic carbon and be careful to explain all acronyms such as TOC (Total Organic Carbon) and REE (Rare Earth Elements).

We fully agree with Referee #3.

Finally, Referee #3 thinks that the thickness of the organic soil horizon should be indicated in the upper panel of Fig. 2. The current colour background indicating the LOI is based on a regression, but the organic topsoil layer is missing because it was hard to make a good regression when including the topsoils. This is most apparent near S22, but it should be possible to add a thin organic topsoil to the figure.

---

## Author Response (AR1)

Below we list the changes in the manuscript that we have done in response to the comments of the three Referees. For a full explanation of our response to the Referees and our decisions we refer to our previous response to the Referees. In some cases their comments were quite general, e.g. the request to be more to the point, and in this case our response is probably best seen in the comparison of the old and the new manuscript below.

**Referee #1:**

**General comments:**

The abstract has been rewritten following the suggestions mainly of Referee #1 (see below). The manuscript has been shortened by removing certain passages and sentences. We have also tried to avoid repeating the impact of riparian soils and the enrichment there too much and made clarifications where Referees have pointed out that it is needed (see details below). In some cases entire passages have been rewritten for improved clarity.

Small overall comments:

Two figures were removed, as suggested by Referee #1.

Referee #1 thought that the title was a bit long. We have now proposed a shorter title.

It is not easy to find datasets to compare our data with, but Referee #1 especially pointed out the data published by Lidman et al. (2014). We have made a comparison to that data, which can be seen in Fig. 6. It is not obvious to us how further comparisons between these two datasets could be made.

**Specific comments:**

1. The reasons for investigating a large group of element has been stated in the abstract: i) many of the investigated elements have not previously been investigated in riparian soils, ii) inclusion of a wide range of elements with contrasting biogeochemical properties makes it easier to see what processes that are important in the riparian zone.
2. We now tried to emphasize the role of TOC and organic matter in the abstract.
3. The abstract has been entirely rewritten following the comments of the three Referees. The two final sentences have been removed.
4. Lines 3-5 state that the riparian zone can affect water quality. Lines 11-13 were meant to state that small streams are important of the runoff generation even in large catchments and that they therefore are important to study. This has been clarified. Lines 13-15 have been removed and the reference has been moved to line 3-5.
5. Lines 23-25 were deleted.
6. We have added examples of why some of the unexplored elements are important, but it would be a very long description if we were to go through each of the 32 elements. Furthermore, the purpose was not to study each of these elements in detail, but to make a broader study of the chemistry of the riparian zone.

7.  We have tried to emphasize that the main reason for using a multi-element approach was to obtain a more over-arching understanding of the riparian zone and the processes that are happening there. We have added a reference to a previous study, where it was possible to predict the behavior of different elements based on their fundamental chemical properties.

8.  The flow pathways were identified using hydrological tracers. This has been added to the manuscript. An overview of the transect is shown in Fig. 1. A cross section of the transect is shown in Fig. 2, where the flow pathways are illustrated in the bottom subplot.

9.  More information on the soils has been added with a more detailed description of where the lysimeters are located. However, because S22 is the only soil profile with distinct soil horizons, S22 is described more thoroughly than S12 and S4.

10. Information about the depth of the organic layer in S22 has been added. There are soil maps of the area, but since all soils in the catchment are classified as till, they are not very useful. Apparently, the organic riparian soils have not been considered to be important. I have colleagues, who are trying to develop models that can predict the width of the riparian zone, but so far we have no verified map that can be published.

11. The information about the porosity has been moved to where the differences in hydraulic conductivity are discussed. That should make it clearer why the porosity is discussed. Information about the sampling depth was added the figure caption. It has also been added that the median and quartiles refer to the measurements that were done every 20 m along the stream that drains the investigated transect.

12. The site description has been changed so that it now starts with more general information and then describes the transect, as suggested by Referee #1.

13. A cross section of the transect is shown in Fig. 2. The bottom subplot illustrates the relationship between discharge and groundwater levels. We have added a reference to Fig. 2 in the second paragraph of 2.2, which Referee #1 was referring to, and stated that it shows the groundwater levels in the transect at different discharge.

14. Given that one of the major concerns of the Referees was that the manuscript should be shortened we have not proceeded with an ANOVA of the vertical trends. Since we do not know how to generalize those results, we fear that it would results in a lengthy discussion. As illustrated by Fig. 2, the main trend in the dataset is the lateral variation so in order to keep the manuscript more concise we believe that it is best to focus on that trend.

15. Figs. S6 and S7 have been removed, as suggested.

16. Table 1 has been updated with the requested information.

17. Since one of the major objectives of the revision was to shorten the manuscript and if possible reduce the number of figures, we have left Fig. S4 in the supplementary material. It is an interesting figure, but we feel that in comparison with the other figures in the manuscript it is less important because it only illustrates a single element instead of a more general trend.

18. The title of section 3.6 has been changed to "the role of riparian soils in the boreal landscape", since Referee #1 thought that the original title was misleading. In addition, parts of the section has been removed or rewritten. We believe that the text now is more to the point and better justifies the title.

19. "More or less" has been removed. Throughout the manuscript we have also tried to use more precise formulations.

*Technical corrections:*

T1: The justification of the names has been removed as requested.

T2: The error has been corrected.

T3: As far as we can see, Fig. S1 is first introduced on the line before Fig. S2 is introduced so it should be correct (p. 4, line 22).

T4: Loss on ignition (LOI) was added to the caption of Fig. S1.

T5: The sentence has been moved to the beginning of the paragraph and rewritten.

5  T6: The passage about pH was shortened so the mistake in question has been removed.

**Referee #2:**

The presentation of the data has been revised. Many parts of the manuscript have been removed or rewritten as shown below. In particular we have tried to make the text less repetitive and more to the point, as requested by all Referees. This should be

10  apparent from the comments in this document, but also from the comparison of the old and the new version of the manuscript.

**Specific comments:**

Page 1, line 9: The reference to lakes has been removed.

15  Page 1, line 10: "More or less" has been removed.

Page 1, line 21: The abstract has been rewritten. We have tried to clearly state problem and briefly describe how the measurements were done. Information about the site has been added to the abstract.

Page 2, line 5: These lines have been rewritten. We have tried now tried to emphasize that the riparian zone is important both for the aquatic systems (water quality) and for the terrestrial systems (fate of nutrients, pollutants etc.).

20  Page 3, line 5: This part of the manuscript has been rewritten. We have now tried to be clearer about the background and the purpose of the study.

**Material and methods**

Site description: The site description has been changed following the suggestion of Referee #2, starting with a general

25  description of the area and then a more detailed descript of the transect.

Fig. 1: It is clearly stated in the manuscript that the study was conducted in one of the subcatchments of the Krycklan catchment (as shown in Fig. 1).The nested approach often used in the Krycklan Catchment Study is, however, not relevant for this study, since it was conducted in one single catchment. Since the manuscript was criticized for having too many figures, it seems hard to motivate the inclusion of a map of the entire Krycklan catchment, in which the subcatchments in

30  question hardly would be visible. (It is one of the smallest ones.) Local soil maps classify all soils as till so they can unfortunately not be used to distinguish or illustrate the riparian zone. Therefore, we cannot supply such a map.

Page 3, line 18: We have now written that the direction of the groundwater was determined using hydrological tracers.

Page 3, line 25: A reference to Fig. S2 has been included as suggested. We also added a reference to Grabs et al., who made a broader investigation of organic matter in riparian soils in Krycklan.

Page 3, line 26: As we replied in our previous comments, mor is a type of humus, which is typical for coniferous forests. It now says "so-called mor" to emphasize that it refers to the thin organic layered discussed in the same sentence. We have checked the term, and it can easily be googled by readers who want to know more about it, so we chose to keep it in the revised manuscript.

Page 3, line 32: The sentence has been rephrased.

Page 4, line 2: A reference is given to Fig. S2, in which the LOI is given.

Page 4, line 31: The sampling has been described in more detail in "2.2. Sampling and analyses". It has also been explicitly stated that water was sampled both from the saturated and from the unsaturated zone.

Page 5 (?), line 13: We have now stated that the loggers, which normally are used to monitor the groundwater levels were out of order during the sampling period. We have also explicitly stated the correlations between discharge and groundwater levels are based on previous research and that the groundwater levels were estimated based on the discharge data.

Page 5, line 23: The relationship was established by one of the co-authors in a previous study, but the results have not been published. Hence, Fig. S3 is the proper reference.

Page 6, line 5: The reference to the definition was already given in the manuscript, but now we have also explained more explicitly what $\Omega$ means.

Page 6, line 6: We have added information about the PCA: the parameters were scaled and centered, but no rotation was used. We have also added that all statistical analyses except the regression in Fig. S4 were done using average concentrations.

Page 6, line 10: Following the recommendations of Referee #3 the section about concentrations in biota has been substantially shortened. Nevertheless, we have tried to properly introduce the questions that these measurements were meant to address, namely the concerns that the enrichment of many potentially toxic elements in the riparian zone also might lead to elevated concentrations in riparian biota. Because the strong enrichment of many potentially toxic elements in the riparian zone was not known previously, we are not aware of previous publications that have addressed this question so no such references have been added.

General comment: We have been careful to cite previous publications whenever we refer to previous measurements or previous conclusions concerning the transect. The results of this study are strictly speaking the numbers presented in Tables 1 and S4. However, an important part of the analysis of this data is to put it into the context of previous research and to try to combine all that is known about this system into a coherent picture. Some of the conclusions we present may therefore rely on data both from this study and data from previous studies combined. The manuscript does not have separate sections for results and discussion, as requested by Referee #2. We found that disposition unsuitable for this manuscript because some of the measurements, for instance the effect on biota, rely heavily on the conclusions from other measurements, e.g. the strong

enrichment in the riparian soil water. Therefore, we would prefer to keep the manuscript with combined section for Results and Discussion.

Table 1: We have added the requested information to Table 1. As for the temporal trends, we believe that it goes beyond the scope of this manuscript, where the idea was to give an overview of the data. We cannot see how we possibly could cover that much data in single manuscript because we have so far not found any way to generalize the temporal trends in a similar manner as the spatial trends.

Fig. 2: The bottom subplot shows the approximate groundwater levels at different discharge exactly as stated in the original manuscript – not the hydraulic condictivity. We have added that this is based on Laudon et al. (2004) in the figure caption.

Page 7, line 3: There was an error here, but the entire passage has been rewritten and shortened.

Page 7, line 8: The observed enrichment of Fe in riparian soil water is a result of this study. However, these results are discussed in the light of previous studies on Fe in this transect and in the Krycklan area. We have rewritten much of the passage about Fe so it should be clearer how these previous results relate to our results. One important conclusion is that the riparian soils are responsible for the absence of Fe colloids in the headwater streams.

Page 7, line 19: We have removed the information about the nine principal components and only focused on the two first ones, which later are shown in the biplot. As we make no further analyses using the principle components, we have not used the Kaiser criterion. Since one of the main messages of the Referees was that we should shorten the manuscript, we have not proceeded any further with the PCA.

Page 7, line 25: As we see it, the role of the PCA is to justify the focus on the lateral trends in the transect. One could possibly argue that this is obvious given the results, but since we make a substantial simplification by just looking at the ratio between the average concentrations in S4 and S22, respectively, we believe that it is a good idea to justify this statistically. The biplot illustrates this is a good way. Instead of focusing more on the PCA, we believe that it is better to focus on the more the mechanistic explanations that are developed in the following sections of the manuscript.

Page 13, line 1: This chapter has been shortened and partly rewritten, trying to avoid repetitions and being more to-the-point, as requested by all Referees. We bring up the questions that Referee #2 mentions in the discussion, but as we said in our previous comments, we do not think that those questions can be answered based on the data we present in this manuscript. Therefore, we have to leave these questions about weathering and transport unanswered for future projects.

**Referee #3:**

Uncertainties in pH:

We state in the manuscript that the average prediction error is 0.24 pH units. The relationship between the modelled and the measured pH is illustrated in Fig. S4. pH was not included in any statistical analyses such as the PCA, which now also is

stated in the manuscript. The passage about pH has been shortened given the limitations of modelled values, as suggested by Referee #3.

Biological uptake:
Referee #3 noted that that it was incorrect to refer to "uptake in biota". In the revised manuscript this expression has been removed. Instead we refer to "concentration in biota", which is what we measured. As suggested by Referee #3, the section has been significantly shortened (ca. 40%), and much of the text has been rewritten. We have also tried to explain more clearly in the revised manuscript why concentrations in biota were measured.

Concerning the organic content of the soils the LOI (loss on ignition) for the three soil profiles is shown in Fig. S1. The variation in organic matter along the stream is further illustrated in Fig. S2.

Acronymns:
The acronym REE has been explained in the revised in the manuscript. The acronym TOC is explained in both the abstract and the manuscript. The explanations were added at the first mentioning of these acronyms, not necessarily those pointed out by Referee #3.

Organic content:
All mentioning of "organic content" have been replaced by "content of organic matter" or a similar expression.

**From soil water to surface water – how the riparian zone controls element transport  from a boreal forest to a stream**

Fredrik Lidman[1], Åsa Boily[1], Hjalmar Laudon[1], Stephan J. Köhler[2]

[1]Department of Forest Ecology and Management, Swedish University of Agricultural Sciences, Umeå, SE-901 83, Sweden
[2]Department of Aquatic Sciences and Assessment, Swedish University of Agricultural Sciences, Uppsala, P.O. Box 7050, SE-750 07, Sweden

*Correspondence to*: Fredrik Lidman (fredrik.lidman@slu.se)

**Abstract.** ~~The riparian zone is the narrow strip of land, which lines lakes and watercourses. In the boreal region the riparian zone of headwaters often tends to be wetland like with high concentrations of organic matter, low pH and more or less reducing conditions. This means that riparian soils in many respects are different from the podzols and other types of mineral soils that dominate the boreal landscape. In this study a large number of major and trace elements (Al, As, B, Ba, Ca, Cd, Cl, Co, Cr, Cs, Cu, Fe, K, La, Li, Mg, Mn, Na, Ni, Pb, Rb, Se, Si, Sr, Th, Ti, U, V, Zn, Zr) and other parameters such as sulphate, total organic carbon (TOC) and pH were analysed in groundwater and soil water in a boreal hillslope. The objective was to investigate how the chemistry changes as the groundwater passes through the riparian zone and enters the stream.~~ Boreal headwaters are often lined by strips of highly organic soils, which are the last terrestrial environment to leave an imprint on discharging groundwater before it enters a stream. Because these riparian soils are so different from the podzol soils that dominate much of the boreal landscape, they are known to have a major impact on the biogeochemistry of important elements such as C, N, P and Fe and the transfer of these elements from terrestrial to aquatic ecosystems. For most elements, however, the role of the riparian zone has remained unclear, although it should be expected that the mobility of many elements are affected by changes in, for example, pH, redox potential and concentration of organic carbon as they are transported through the riparian zone. Therefore, soil water and groundwater was sampled at different depths along a 22 m hillslope transect in the Krycklan catchment in northern Sweden using soil lysimeters and analysed for large number of major and trace elements (Al, As, B, Ba, Ca, Cd, Cl, Co, Cr, Cs, Cu, Fe, K, La, Li, Mg, Mn, Na, Ni, Pb, Rb, Se, Si, Sr, Th, Ti, U, V, Zn, Zr) and other parameters such as sulphate and total organic carbon (TOC). The results showed that the concentrations of most investigated elements increased substantially (up to 60 times) as the water flowed from the uphill mineral soils and into the riparian zone, largely as a result of higher TOC concentrations. The stream water concentrations of these elements were typically somewhat lower than in the riparian zone, but still considerably higher than in the uphill mineral soils, which suggests that riparian soils have a decisive impact on the water quality of boreal streams. The degree of enrichment in the riparian zone for different elements could be linked to the affinity for organic matter, indicating that the pattern with strongly elevated concentrations in riparian soils is typical for organophilic substances. One likely explanation is that the solubility of many organophilic elements increases as a result of the higher concentrations of TOC in the riparian

zone. Elements with low or modest affinity for organic matter (e.g. Na, Cl, K, Mg and Ca) occurred in similar or lower concentrations in the riparian zone. Despite the elevated concentrations of many elements in riparian soil water and groundwater no increase in the concentrations in biota could be observed (bilberry leaves and spruce shoots).

[revised manuscript text omitted]

**Formaterat:** Teckenfärg: Dekorfärg 4

The investigated transect is located ca. 300 m upstream of the catchment outlet and consists of three sampling sites (S4, S12 and S22) located 4 m, 12 m and 22 m, respectively, from the stream as measured along the flow pathway of the groundwater, which was identified using hydrological tracers (Fig. 1). At each sampling site ceramic suction lysimeters were installed in 1995, allowing regular sampling of soil water and groundwater at different depths in the three soil horizons. In this study four depths at S4 and five depths each at S12 and S22 were sampled. Each lysimeter is referred to by the name of sampling site followed by the depth in cm, e.g. S4-45. S22 represents an iron podzol with clearly developed soil horizons. As such it is typical for most of the investigated catchment and in a broader sense also for boreal forests in general. At the surface there is a ca. 8 cm thick organic layer (so-called mor), but apart from that the content of organic matter is low (<0.8% below 10 cm) (**Fel! Hittar inte referenskälla.**S1). The uppermost lysimeter (S22-20) is located at the lower end of the E horizon, which reaches from 8-20 cm. Then there are three lysimeters (S22-35, S22-50 and S22-75) in the B horizon, which has a hardpan around 75 cm. Finally, there is one lysimeter in the C horizon (S22-90), which starts at ca. 80 cm. This lysimeter is also located in the compact basal till with notably lower conductivity (Table S2). As the groundwater approaches the stream the podzol gradually gives way for more organic histosols as a result of the increasingly wet conditions closer to the stream. S4 and S12 can both be considered to represent the riparian zone and lack any distinct soil horizons. The thickness of the organic layer increases from 20-30 cm at S12 to ca. 80 cm at S4 (**Fel! Hittar inte referenskälla.**S1). This accumulation of organic matter in the riparian zone is typical for this and other headwaters in the Krycklan catchment (Fig. S2; Grabs et al., 2012). The slope is not particularly steep with an inclination of ca. 3%, which can be compared to the average inclination of the Västrabäcken catchment, 8.7%. Uphill from S22 the slope continues ca. 100-120 m to the water divide (Figure 1). Previous research has demonstrated that in all three soil profiles (S4, S12 and S22) there is a clear relationship between the groundwater level and stream discharge (Laudon et al., 2004; Seibert et al., 2009). The discharge increases exponentially with rising groundwater levels following the so-called transmissivity feedback mechanism, which suggests that much of the water transport takes place the uppermost saturated soil layers (Fig. 2; Bishop et al., 2011). This is also related to the porosity of the soils, which varies from 36-83% with higher values in more organic soils and, consequently, a decline with depth in all profiles (**Fel! Hittar inte referenskälla.**). Further details on the transect can be found elsewhereprevious publications (Nyberg et al., 2001; Stähli et al., 2001).

**2.2 Sampling and analyses**

Soil water and groundwater (i.e. both from the saturated and the unsaturated zone) was collected from the lysimeters at ten occasions during 2008 by attaching vacuum bottles to the tubes that are connected to each of the ceramic cups in the soils. The investigation period started during winter baseflow conditions in January and ended in October, when the system again had returned to winter baseflow conditions. Samples were collected every month with the exception of June, when two samples were collected, and September, which was omitted. This sampling strategy was assumed to capture the most active period both in hydrological and biogeochemical terms. This means that there are in most cases ten observations from each depth in each profile, but in certain cases there are values missing either because sufficient amounts of water could not be

collected due to soil frost or drought or because the concentrations of some elements in some cases were below detection limit.

In S4 four lysimeters were sampled, reaching from 35-65 cm in depth. This covers the most hydrologically dynamic part of the profile, since S4-35 never was saturated, except possibly briefly in connection with the peak flow of the spring flood, and S4-65 was constantly saturated, except for a few weeks in the middle of the summer. In S12 five lysimeters covering a depth from 20-70 cm were sampled. S12-20 is believed never to have been saturated, while the groundwater level occasionally may have fallen below S12-70 during the driest parts of the summer. In S22 five lysimeters reaching from 20-90 cm were sampled. Except in connection with the peak flow of the spring flood S22-20 is expected to have been above the groundwater table, whereas S22-75 and S22-90 are expected to have been in the saturated zone throughout the entire sampling period. Approximate groundwater levels at different discharge are shown in the  bottom subplot of Fig. 2. As the groundwater loggers were out of order during the sampling period, the groundwater levels were reconstructed using continuous discharge measurements form a nearby station, where a pressure transducer continuously measures the water height at a V-notch weir (Fig. 1; Seibert et al., 2009). Previous research has shown that the groundwater levels in all three soil profiles are correlated to the discharge (Laudon et al., 2004). Hence, the discharge data was used to estimate the groundwater levels using slightly revised regression from Laudon et al. (2004) with $R^2$ values of 0.96 (S4), 0.94 (S12) and 0.87 (S22), respectively (p<0.001).

The chemical analyses included Cl, sulphate, total organic carbon (TOC), Al, As, B, Ba, Be, Ca, Cd, rare earth elements (REEs), Co, Cr, Cs, Cu, Fe, K, Li, Mg, Mn, Na, Ni, Pb, Rb, Se, Si, Sr, Th, Ti, Tl, U, V, Zn and Zr. Because of the large similarities between the REEs only La is presented in this study in order to save space and make graphs and tables clearer and more manageable. The fractionation within the lanthanide series will for this reason have to be addressed elsewhere. pH was not directly measured in the water samples from 2008, but based on previous measurements in this transect (1996-1998) there is a strong relationship between pH on one hand and TOC and Ca on the other (**Fel! Hittar inte referenskälla.**S3), which was used to estimate the pH of the analysed water samples in this study. The average prediction error was 0.24 pH units. Additional analyses of water from the adjacent stream (referred to as C2 or Västrabäcken in previous studies) were taken from the material presented by Lidman et al. (2014).

The soil water and groundwater was sampled by attaching vacuum bottles to the lysimeters for 2-3 days. The lysimeters were carefully installed in 1995 so any effects of disturbance on the soils were expected to have evanesced. The samples were filtered (0.45 μm) and acidified to pH<2 with ultrapure double-distilled $HNO_3$. TOC was then analysed by a Shimadzu TOC-VCPH instrument. Previous research in Krycklan has demonstrated that the amount of particulate organic carbon rarely

exceeds a few percent in these systems so TOC is essentially equal to the dissolved organic carbon (DOC) (Laudon et al., 2011). Anions were measured by ion chromatography (Dionex ICS-90, Sunnyvale, Ca, USA; 4 mm i.d. AG14 and AS14 columns) using a suppressor and conductivity detector. All other elements were analysed by ICP-MS (Perkin-Elmer ELAN 6000).

The speciation of elements in the riparian soil water was calculated using thermodynamic modelling in Visual MINTEQ 3.1 (Gustafsson, 2012). The binding to DOC was modelled using the Stockholm Humic Model as described by Sjostedt et al. (2010). The modelled association to DOC (in %) was then used as an index for the affinity for organic matter (Ω) based on the definition suggested by Lidman et al. (2014) . An element with Ω close to 100% is considered to have strong affinity for organic matter , whereas an element with Ω close to 0% is considered to have low affinity for organic matter. In one regression analysis Ω was transformed using the logit function:

$$logit(\Omega) = log\left(\frac{\Omega}{1-\Omega}\right) \hspace{4cm} (1)$$

All statistical analyses, including the principal component analysis (PCA), were made in R (R Core Team, 2014). In order to remove all missing values Cr and Th were excluded from the PCA of the soil water and groundwater and one value each for U and Zr were interpolated from the two surrounding lysimeters (Table 1). Because pH was modelled it was not included in the PCA. The variables were scaled and centred, but no rotation was used. All statistical analyses were done using average concentrations except in Fig. S4.

The biological uptake in the riparian zone and in the upslope soil, respectively, was tested by collecting fresh bilberry leaves (*Vaccinium myrtillus*) and spruce shoots (*Picea abies*) early in the growing season (11 June 2013). Two grouped samples of each species were collected in the vicinity of the lysimeters at S4 and S22, respectively. A screening of the element concentrations were made using ICP-MS at a commercial laboratory (ALS Global, Luleå, Sweden) following certified standard procedures. We report the results for the 19 elements, which were measured with an accuracy of two of more significant digits.

**3 Results and discussion**

**3.1 Soil water and groundwater chemistry**

For most elements there was a substantial increase in the concentrations along the transect, from the upslope profile (S22) to the riparian profile (S4) (Table 1, **Fel! Hittar inte referenskälla.**). These patterns, which were persistent throughout the

**Formaterat:** Håll inte ihop med nästa

year, are exemplified by TOC and V in Figure 2. In the case of TOC the increase in riparian soil water and groundwater (up to 33 mg L$^{-1}$ on average) was not surprising given that the  content of organic matter in the riparian soils is substantially higher  than the upslope soils, which typically contained only a few mg L$^{-1}$ (**Fel! Hittar inte referenskälla.**S1). This  trend is typical for the riparian soils of many small boreal streams (Grabs et al., 2012). V followed that same trend as TOC with substantially higher (nearly two orders of magnitude) concentrations in the riparian zone. However, there were also elements, whose concentrations did not increase markedly in the riparian zone. One such example is Na, which occurred in relatively stable concentrations around 2 mg L$^{-1}$  (Fig. 2).

~~pH was estimated to be on average 4.5 in S4, ranging from 4.2-4.7 with slightly higher values in deeper horizons: on average 4.3 in S4-35 as compared to 4.6 in S4-64. In S12 the pH was on average 5.0, covering an interval from 4.4 to 5.2. Again there was a tendency of higher pH in deeper horizons. The average pH in S22 was only slightly higher than in S12, 5.1, but there was a notable difference of more than one pH unit between S22-90, which had an average pH of 5.9, and the upper horizons, which had an average of 4.8. Due to the stable pH in all profiles of S22 this also coincides with the observed range in pH, 4.8-5.9. Hence, the pH gradient experienced byunits aloneand even more soconcentration (().Detailed studies of the Fe speciation using X-ray absorption spectroscopy (XAS) have revealed considerable complexity with simultaneous presence of both Fe2+ and Fe3+ throughout the transect~~ Fe was also strongly enriched in riparian soil water with average concentrations ranging from 5.3 µg L$^{-1}$ at the bottom of the E horizon (S22-20) to 1500 µg L$^{-1}$ in S4-45. Previous studies using X-ray absorption spectroscopy (XAS) have demonstrated that both Fe$^{2+}$ and Fe$^{3+}$ are present throughout the transect, but in the riparian zone the Fe speciation was completely dominated by organic Fe complexes (Sundman et al., 2014). The dominance of organically bound Fe in the riparian zone is consistent with the absence of Fe colloids in the adjacent stream (Neubauer et al., 2013). This observation is important because Fe colloids are an important vector for many major and trace elements in boreal rivers (Gustafsson et al., 2000; Dahlqvist et al., 2007)

of the riparian zone apparently destabilised or blocked these precipitates from reaching the stream water. Only as the significance of the riparian zone decreased with increasing catchment area, e.g. due to less organic riparian zones and a larger influence of deep groundwater. If no Fe colloids are present, however, organic colloids are likely to play the lead role in transporting more insoluble elements through the riparian zone. Only as the streams become larger, Fe colloids will gradually appear in the stream water (Neubauer et al., 2013). A prerequisite for this is that pH increases with increasing catchment area, while the TOC concentration decreases, which causes both Fe and Al to precipitate (Kohler et al., 2014; Lidman et al., 2014). This is in turn related to less organic matter in the riparian soils of larger streams and an increasing influence from deeper groundwater, which entirely may bypass the riparian soils (Peralta-Tapia et al., 2015; Lidman et al., 2016), will these Fe colloids start to appear in the stream water  (Neubauer et al., 2013)..

**3.2 Principal component analysis**

[revised manuscript text omitted]
 content of organic contentcarbon (around the 3rd quartile) than most reaches of the stream (**Fel! Hittar inte referenskälla.** and S2). If the interpretation that the organic matter is controlling the transport of these elements is correct, that could explain why the studied riparian zone exhibited higher concentrations than the stream. Differences in the organic content of organic matter in the riparian zone along the stream is known to derive from variations in the topography and the groundwater flow pathways  (Tiwari et al., 2016). This hypothesis is also supported by the fact that the TOC concentrations in the riparian zone also were higher than in the stream (**Fel! Hittar inte referenskälla.**).

There were only three elements, Ca, Mg and K, which displayed the opposite pattern: higher concentrations in the stream water (C2) than in the riparian soil water (S4) (**Fel! Hittar inte referenskälla.**). These three elements also stood out in the measurements in the transect because they occurred in higher concentrations in the mineral soil (S22) than in the riparian soil (S4), mainly due to the high concentrations in the deepest lysimeter in S22 (Figure 4). In the case of Ca and Mg, the agreement between the mineral soil water (S22) and the stream water (C2) was good, whereas the K concentration in the stream water was more than twice as high as in the mineral soil water. K and, possibly, to some extent Rb were the only examples of elements with notably higher concentrations in the stream water than in the three soil profiles. The higher concentrations of base cations like K, Ca and Mg in the stream water could indicate a contribution of deeper groundwater to the stream, but at the same time related weathering-products such as Sr, Na, Rb, Cs and Si all showed relatively uniform concentrations throughout the transect and the stream.  It is possible that the differences again are causecaused by longitudinal heterogeneities along the stream.

3.5 Biological uptake

It has been shown that the 3.5 A hotpot for transfer of pollutants to biota?

Higher species richness in the riparian zone of stream reaches with high groundwater discharge.  is a clear sign that the ecological community can benefit from the flow of nutrients into the riparian zone  (Kuglerová et al., 2013).  However, not only nutrients, but also a large range of potentially toxic substances, tend to accumulate in the riparian zone . It is therefore pertinent to ask whether there also is an elevated accumulation in riparian biota, which could constitute an important exposure pathway for

hazardous substances. The screening of bilberry leaves and spruce shoots from the riparian forest (near S4) and the uphill forest (near S22), respectively, showed no clear relationship between the concentrations in soil water and the concentrations in biota (**Fel! Hittar inte referenskälla.**S4; Figure 7**Fel! Hittar inte referenskälla.**). Also when considering the more unreliable results for elements, which were only reported with one significant digit (e.g. Sb, Pb, Cr, La, Ti, V and Zr), there were no clear signs of elevated concentrations in riparian biota. (Possible exceptions were, however, As, Co and Mo.) ~~Combined these results suggest that the biological uptake either in general is actively regulated by the plants or that the elements are present as species with low bioavailability. The two possibilities do not exclude each other and may vary from element to element, but according to the discussion above and the modelling of the chemical speciation one should expect a large portion of the elements with elevated concentrations in the riparian soil water to be associated with TOC (Fig. 5). For a wide range of metals it has been shown that their(Van Sprang et al., 2015)Although there may be differences between species, the general effects on the bioavailability are likely to be similar also for spruce, bilberry and many other plants. Hence, the fairly constant concentrations in biota could be explained by the fact that the concentrations of the most bioavailable chemical species do not differ much between the inorganic uphill soils and the riparian zone. Instead, the enrichment occurs mainly by increasing the amount of the organically bound fractions, which have low bioavailability.~~ In addition, it is possible that the plants are able to regulate the uptake and/or disposal of certain elements.

**Formaterat:** Engelska (Storbritannien)

 While plants growing in the riparian zone generally did not have higher concentrations of elements  that accumulate in riparian soil water, there  were two notable  exceptions: Cs and Mn. In the case of Cs  2-3 times higher  concentrations in  uphill soil,   soil water led to 4-5 times higher.  concentrations in biota. Since Cs is considered to have low affinity for organic matter, the  elevated  concentrations  are probably not directly linked to TOC (Fan et al., 2014).  Instead, factors such as lower amount of mineral surfaces and lower pH  could allow higher Cs concentration in the riparian zone.  concentrations of more bioavailable Cs ions in the riparian  soil water, leading to elevated concentrations in biota.  In the case of Mn the concentrations in riparian biota were lower, which possibly could be related to its redox  chemistry (Fig. 7).

**3.6 The  role of  riparian  soils in the boreal landscape**

The perhaps most important observation in this study was the large differences between the riparian soils and the uphill forest soils . As discussed above, these differences have previously been known for a  number of elements, but judging from this study the enrichment in riparian soil water is more the  rule  than  the exception (Figure 4).  Since the enrichment in  riparian soil water  could be linked to the affinity for organic matter (Figure 5). similar enrichment should be expected for many other metals and organophilic substances in general, e.g. organic pollutants (Bergknut et al., 2011). A crucial question then is unavoidably how representative the investigated transect would be over at larger spatial scales. Even in a small catchment like this one (0.12 km$^2$) the chemistry of the draining stream not agree completely with either the water of the riparian zone or that of the uphill mineral soil (**Fel! Hittar inte referenskälla.**). Given the heterogeneity and complexity of the natural landscape it is doubtful whether it at all would be possible to find something like a representative transect. In all catchments there is likely to be substantial longitudinal and transverse variation in state factors such as topography, grain size distribution and mineralogy, which in turn gives rise to a heterogeneity in hydrology and biogeochemistry (Ledesma et al., 2015). These differences are likely to only increase when trying to compare different catchments to one another. Nevertheless, it was clear that the composition of the stream water had been significantly influenced by the riparian zone, e.g. by gaining substantially higher concentrations of TOC and organophilic elements

~~Since this study focused on a single hillslope transect, a key question is unavoidably how representative the results are. Needless to say, factors such as topography, hydrology, soil composition etc. will vary from catchment to catchment so there is little reason to expect the exact numbers, e.g. the degree of enrichment of organophilic elements in riparian soils, to repeat themselves. On a more conceptual level, however,the gradient represented by this transect from relatively dry organic-poor mineral soils in uphill locations to wetter organic soils near the streams is surely not untypical for the boreal landscape (Sauer et al., 2007; Grabs et al., 2012). Despite their size, small streams and headwaters are fundamental for generating runoff in the boreal landscape (Bishop et al., 2008). If the water quality of these streams and the transfer of TOC and associated elements are controlled by riparian soils, as supported by this study, there are also good reasons to believe that the effects of the riparian soils have implications at much larger scales than the investigated transect (Laudon et al., 2011; Ledesma et al., 2015). The results clearly demonstrated that the riparian zone had a profound impact on the stream draining the transect (Fig. 6), and previous research in the Krycklan catchment has shown that the water chemistry of this stream (C2) in many respects is typical for forested catchments in the area despite its small size~~ (Cory et al., 2006; Björkvald et al., 2008; Lidman et al., 2012; Köhler et al., 2014; Lidman et al., 2014). Despite their size, small streams and headwaters are important for generating runoff in the boreal landscape, even in large rivers (Bishop et al., 2008). Therefore, on a more conceptual the gradient represented by this transect from relatively dry organic-poor mineral soils in uphill locations to wetter organic soils near the streams is surely not untypical for the boreal landscape (Sauer et al., 2007; Grabs et al., 2012). If the water quality of these streams and the transfer of TOC and associated elements largely are controlled by riparian soils, as suggeseted by this study, there are also good reasons to believe that the effects of the riparian soils have implications at much larger scales than the investigated transect (Laudon et al., 2011; Ledesma et al., 2015). These effects include affecting the balance between organic colloids and Fe colloids, lowering pH and elevating the concentration of TOC and many organophilic elements (Lyven et al., 2003; Neubauer et al., 2013; Köhler et al., 2014;  Ledesma et al., 2015).

~~Arguably, the chemistry of the stream draining the investigated transect did not agree completely with either the water of the riparian zone or that of the uphill mineral soil (Fig. 6). However, given the heterogeneity and complexity of the natural landscape it is doubtful whether it at all would be possible to find something like a representative transect, even in a relatively small catchment like this one (0.12 km²). In all catchments there is likely to be substantial longitudinal and transverse variation in state factors such as topography, grain size distribution and mineralogy, which in turn gives rise to a heterogeneity in hydrology and biogeochemistry (Ledesma et al., 2015).Nevertheless, it was clear that the composition of the stream water had been significantly influenced by the riparian zone, e.g. by gaining substantially higher concentrations of TOC and organophilic elements in general (Fig.~~

Formaterat: Engelska (USA)

Ändrad fältkod

4). For an element like C, which ultimately is derived from the atmosphere, it is not hard to see how riparian soils can become a major source of TOC to the streams (Fiebig et al., 1990; Agren et al., 2008; Lyon et al., 2011). More intriguing are the associated elements, which in a similar manner as TOC have a major source in the riparian zone. These elements, however, are largely derived from weathering of local soils, but it is unclear where in the catchment this weathering takes place and what the long-term role riparian soils play in the boreal landscape. One hypothesis is 
[revised manuscript text omitted]